# Convergence of cMyc and β-catenin on Tcf7l1 enables endoderm specification

Gillian Morrison[1,†,*], Roberta Scognamiglio[2,3], Andreas Trumpp[2,3] & Austin Smith[1,4,**]

## Abstract

The molecular machinery that directs formation of definitive endoderm from pluripotent stem cells is not well understood. Wnt/β-catenin and Nodal signalling have been implicated, but the requirements for lineage specification remain incompletely defined. Here, we demonstrate a potent effect of inhibiting glycogen synthase kinase 3 (GSK3) on definitive endoderm production. We find that downstream of GSK3 inhibition, elevated cMyc and β-catenin act in parallel to reduce transcription and DNA binding, respectively, of the transcriptional repressor Tcf7l1. Tcf7l1 represses FoxA2, a pioneer factor for endoderm specification. Deletion of Tcf7l1 is sufficient to allow upregulation of FoxA2 in the presence of Activin. In wild-type cells, cMyc contributes by reducing *Tcf7l1* mRNA, while β-catenin acts on Tcf7l1 protein. GSK3 inhibition is further required for consolidation of endodermal fate via upregulation of Sox17, highlighting sequential roles for Wnt signalling. The identification of a cMyc/β-catenin-Tcf7l1-FoxA2 axis reveals a de-repression mechanism underlying endoderm induction that may be recapitulated in other developmental and pathological contexts.

**Keywords** embryonic stem cells; endoderm; FoxA2; Myc; Tcf7l1
**Subject Categories** Development & Differentiation
The EMBO Journal (2016) 35: 356–368

## Introduction

Nodal/Activin signalling is required for endoderm specification in the embryo (Brennan *et al*, 2001; Vincent *et al*, 2003) and steering differentiation of pluripotent stem cells towards definitive endoderm (Kubo *et al*, 2004; Gadue *et al*, 2006; Morrison *et al*, 2008). Canonical Wnt signalling is also implicated in the formation of endoderm and mesoderm during early development (Liu *et al*, 1999; Huelsken *et al*, 2000; Tsakiridis *et al*, 2014) and can enhance expression of primitive streak-associated genes in differentiating embryonic stem (ES) cells (Gadue *et al*, 2006; ten Berge *et al*, 2008).

The transcription factor FoxA2 is the earliest known marker of definitive endoderm. Its upregulation at the onset of gastrulation is considered to represent a crucial step in commitment to definitive endoderm identity (Levinson-Dushnik & Benvenisty, 1997; Burtscher & Lickert, 2009). Surprisingly, we have a limited understanding of the signalling processes that activate FoxA2 expression and initiate molecular specification of definitive endoderm.

Mouse ES cells are pluripotent cell lines with a capacity to differentiate into all cell types of the developing and adult organism (Bradley *et al*, 1984; Smith, 2001). The heterogeneity of ES cells maintained in the presence of serum (Hayashi *et al*, 2008; Silva & Smith, 2008) can be a confounding factor when attempting to optimise lineage specification and delineate initial molecular and cellular transitions. ES cells can, however, be maintained as a substantially homogeneous population of naïve stem cells when cultured without serum in the presence of inhibitors of both MEK/Erk signalling and glycogen synthase kinase-3; termed "2i" (Ying *et al*, 2008; Wray *et al*, 2010). Upon release from the naïve ground state conditions, ES cells readily undergo developmental progression and lineage specification (Hayashi *et al*, 2011; Marks *et al*, 2012; Betschinger *et al*, 2013; Buecker *et al*, 2014; Kalkan & Smith, 2014; Yang *et al*, 2014).

Here, we exploit the experimental tractability of ground state ES cell differentiation to explore molecular events that link signalling processes with transcriptional changes during acquisition of definitive endoderm identity. In particular, we focus on pivotal events downstream of inhibition of GSK3.

## Results

### Suppression of GSK3 activity during exit from naïve pluripotency promotes endoderm lineage specification

We first examined the response of naïve ES cells to a well-established cocktail of endoderm differentiation inducers (Kubo *et al*, 2004; Gadue *et al*, 2006; McLean *et al*, 2007; Morrison *et al*,

1 Wellcome Trust-Medical Research Council Cambridge Stem Cell Institute, University of Cambridge, Cambridge, UK
2 Division of Stem Cells and Cancer, Deutsches Krebsforschungszentrum (DKFZ), Heidelberg, Germany
3 Heidelberg Institute for Stem Cell Technology and Experimental Medicine (HI-STEM gGmbH), Heidelberg, Germany
4 Department of Biochemistry, University of Cambridge, Cambridge, UK
*Corresponding author. Tel: +44 1316 519500; E-mail: gillian.morrison@ed.ac.uk
**Corresponding author. Tel: +44 1223 760233; E-mail: austin.smith@cscr.cam.ac.uk
†Present address: MRC Centre for Regenerative Medicine, Scottish Centre for Regenerative Medicine Building, The University of Edinburgh, Edinburgh, UK

 

2008)(Activin, FGF4, heparin and PI3K inhibitor) in conjunction with withdrawal from 2i. We detected only a minor increase in the number of cells expressing the endoderm markers CXCR4 and Sox17 after 7 days (Fig 1A). Therefore, Activin is not sufficient to drive naïve pluripotent cells into the endoderm lineage. Previous studies have suggested that endoderm may be increased in response to Wnt signalling or GSK3 inhibition (D'Amour *et al*, 2005; Gadue *et al*, 2006; Bone *et al*, 2011; Loh *et al*, 2014). We therefore applied the GSK3 inhibitor CHIR99021 (CH) in combination with the other factors (Fig EV1). This resulted in a striking and reproducible increase in the proportion of cells positive for CXCR4 from < 15% to ~80% (Fig 1A). A similar result was observed for Sox17 immunostaining (Fig 1A).

Gene expression analysis confirmed that Sox17, Hex and additional endoderm markers were significantly upregulated (Fig 1B). Mesodermal and neural markers were not induced, and ES cell pluripotency markers Oct4 and Nanog were downregulated (Fig 1B). Co-staining for E-cadherin and FoxA2 at day 7 of differentiation provided further evidence of efficient production of endodermal cells in the presence of CH (Fig 1C). Extraembryonic endoderm markers Sox7 and AFP were not expressed and less than ten per cent of cells were positive for PDGFRα at day 7, a marker of both extraembryonic endoderm and mesoderm (Fig EV1). The differentiation process displayed a sequence of gene expression changes that mirror those observed in early mouse development (Arnold & Robertson, 2009) with the primitive streak/early endoderm marker CXCR4 detected at day 2–3 and Sox17 from day 4 (Fig 1D and E).

To investigate which events are affected by GSK3 inhibition in the context of endoderm inductive conditions, a time-course analysis of marker gene expression was performed (Fig EV1). Downregulation of the naïve pluripotency marker Nanog was slightly delayed by GSK3 inhibition consistent with the known effects on self-renewal circuitry (Martello *et al*, 2012). Upregulation of the early primitive streak markers Tbra and Wnt3 at 24–48 h is promoted by GSK3 inhibition, but the most potent effect is on the induction of the key early definitive endoderm marker FoxA2; CH is a requirement for induction of FoxA2 as ES cells exit the naïve state.

By administering pulses of CH treatment, we found that GSK3 inhibition is critical for endoderm differentiation between days 1–5 (Fig EV1). Withdrawal of CH after induction of early primitive streak cells (Tbra/Wnt3 positive) at day 2 was not sufficient for emergence of endoderm cells. Similarly, removal of CH at day 3 following induction of FoxA2 did not allow Sox17 expression. We conclude that the inclusion of GSK3 inhibitor throughout this differentiation regime promotes the successive emergence of early primitive streak-like cells and definitive endoderm.

Five alternative inhibitors with distinct chemical structures yielded a similar production of endoderm as CH (Fig EV1), strongly indicating that the effect is mediated specifically through GSK3 inhibition. Importantly, Activin was essential for appreciable induction of endoderm markers (Fig EV1). Fgf4, heparin and the PI3K inhibitor PI103 were each partially dispensable, but collectively substantially increased the final yield (Fig EV1).

We tested five different ES cell lines, including one derived from NOD mice (a background genetically distinct from conventional

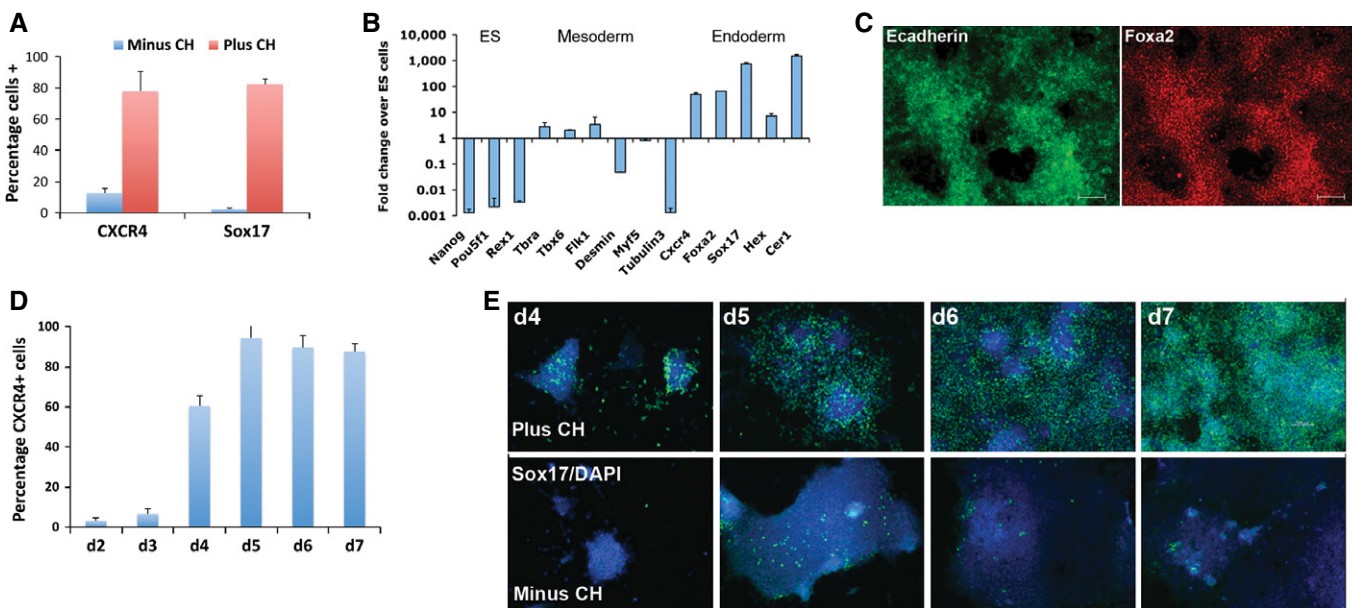

**Figure 1.  Suppression of GSK3 activity promotes endoderm production from naïve ES cells.**

A  Quantification of CXCR4$^+$ cells by flow cytometry and Sox17$^+$ cells by immunostaining at day 7 following differentiation in the presence or absence of 3 μM CH. Average and standard deviation (SD) of eight independent experiments.

B  Gene expression profiling by RT–PCR at day 7 of differentiation. Average and SD of three independent experiments.

C  Immunostaining for E-cadherin (left) and FoxA2 (right) at day 7 of differentiation in the presence of 3 μM CH. Scale bars, 200 μm.

D  Quantification of CXCR4$^+$ by flow cytometry over a 7-day differentiation time course in the presence of 3 μM CH. Average and SD of three independent experiments.

E  Immunostaining for Sox17 at day 4 to day 7 of differentiation in the continuous presence or absence of 3 μM CH. Scale bars, 200 μm.

strain 129 ES cells). All produced > 70% Sox17$^+$ cells on day 7 of differentiation (Fig EV1), demonstrating reproducibility of this protocol. Induced endodermal cells were able to differentiate further on application of established protocols (D'Amour et al, 2005; Morrison et al, 2008) for generation of pancreatic and hepatic progenitors (Fig EV1).

Collectively, these findings demonstrate that inhibition of GSK3 facilitates formation of definitive endoderm from naïve ES cells in response to Nodal/Activin.

### GSK3 inhibition reduces expression of the transcriptional repressor Tcf7l1 (Tcf3)

We investigated candidate pathways through which inhibition of GSK3 might relieve a roadblock to definitive endoderm. GSK3 inhibition has been demonstrated to reduce activity of the transcriptional repressor Tcf7l1 (also known as Tcf3) in undifferentiated ES cells (Wray et al, 2011; Yi et al, 2011; Martello et al, 2012). We reasoned that under differentiation conditions, CH might facilitate endodermal specification by de-repression of additional Tcf7l1 targets. We examined expression of Tcf7l1 during the early phase of endodermal differentiation and found that Tcf7l1 mRNA was reduced approximately twofold in the presence of CH (Fig 2A). Levels of Tcf7l1 protein were even more severely affected (Fig 2B). The decrease in Tcf7l1 preceded an increase in expression of the earliest endodermal gene FoxA2 (Fig 2C).

### GSK3 inhibition reduces binding of Tcf7l1 to the FoxA2 gene locus

Using the CODEX compendium of ES cell ChIP-Seq data sets (Sánchez-Castillo et al, 2015), we surveyed a range of candidate regulators of the endoderm lineage to search for potential direct transcriptional targets of Tcf7l1. Binding peaks were identified in the region of several genes associated with early endodermal differentiation, including FoxA2, Nodal and Eomesodermin (Fig 2D). A single Tcf7l1 binding peak identified at the FoxA2 locus is located within a 250-bp region that is highly conserved within mammals (Fig EV2). Two TCF/LEF consensus DNA-binding motifs (van Beest et al, 2000) are present within this binding peak (Fig EV2).

ChIP-PCR on wild-type and Tcf7l1 null ES cells confirmed specific binding of Tcf7l1 to the FoxA2 locus at similar levels to that observed for Klf2, a previously validated target (Martello et al, 2012). Significant binding was also validated at Eomes (Fig 2E). Strikingly, inclusion of CH during differentiation significantly reduced Tcf7l1 binding to the FoxA2 locus at day 3 compared to cells without CH (Fig 2F). Diminished Tcf7l1 binding correlates with the early increase in FoxA2 expression (Fig 2C). We surmise that removal of direct transcriptional repression of FoxA2 by Tcf7l1 may be an initiating molecular event in the emergence of endodermal identity and can be triggered by pharmacological inhibitors of GSK3.

### Tcf7l1 null ES cells do not require CH to initiate endodermal differentiation

To examine this hypothesis, we tested whether elimination of Tcf7l1 could replace the requirement for CH in the induction of endoderm. Tcf7l1 null cells exhibit a general delay in entry into differentiation

due to sustained expression of the pluripotency gene regulatory network (Pereira et al, 2006; Guo et al, 2011; Martello et al, 2012; Betschinger et al, 2013). Consistent with this, FoxA2 was not fully upregulated until day 4, approximately 24 h later than in wild-type control cells (Fig 3A). Strikingly, however, at day 4 of differentiation, the expression of FoxA2 in Tcf7l1 null cells without CH was equal to or higher than that in wild-type cells with CH (Fig 3B). Genetic ablation of Tcf7l1 therefore renders FoxA2 induction independent of GSK3.

Eomesodermin and Cerl1, markers of nascent endoderm, were also induced in Tcf7l1 null cells without CH (Fig 3B). FoxA2 protein was also detected (Fig 3C). However, although Sox17 and other endodermal marker genes that are upregulated later than FoxA2 (e.g. CXCR4 and Hex) were expressed at higher levels in Tcf7l1 null cells than in control cells without CH, levels did not reach those in cells treated with CH (Fig 3D). Thus, absence of Tcf7l1 removes the requirement for inhibition of GSK3 for initial, but not later, stages of endodermal lineage specification.

### Induction of FoxA2 as cells exit pluripotency can replace GSK3 inhibition in enabling endoderm specification

The preceding data indicate that the critical response to CH treatment during early differentiation is to relieve Tcf7l1 repression of FoxA2. A prediction of such a model is that forced expression of FoxA2 would be sufficient to trigger endodermal specification in the absence of CH. To test this, we generated two independent ES cell lines containing a doxycycline-inducible FoxA2 transgene. Addition of doxycycline (DOX) increased FoxA2 mRNA and protein, but was insufficient to impose endodermal differentiation under self-renewal conditions (2i) when other inputs are blocked (Fig 4A and B). Under differentiation permissive conditions, addition of DOX resulted in expression of the endoderm marker CXCR4 by day 3 without requirement for CH (Fig 4C). We noted, however, that few Sox17-positive cells were obtained in the continued absence of CH (Fig 4D). We therefore added CH at day 4, following induction of FoxA2. Equivalent numbers of Sox17 cells were obtained as from cultures in CH from day 0 to 7 (Fig 4D and E). Importantly, Sox17 could only be induced by late addition of CH if this was preceded by early FoxA2 induction; FoxA2 induction after day 3 resulted in very few Sox17 cells. These observations are consistent with sequential and mechanistically distinct roles for CH; early FoxA2 induction via de-repression of Tcf7l1, and subsequent upregulation of Sox17 to consolidate definitive endoderm identity, likely involving β-catenin-mediated transcriptional activation via other Lef/Tcf family members.

### Downregulation of Tcf7l1 occurs independently of Wnt/β-catenin

We examined substrates of GSK3 that might explain the downregulation of Tcf7l1. GSK3 has pleotropic roles as a negative regulator of diverse pathways and processes (Doble, 2003) and is a well-established regulatory component of the canonical Wnt/β-catenin signalling pathway. A direct protein–protein interaction between Tcf7l1 and β-catenin has been documented (Hikasa & Sokol, 2011; Shy et al, 2013). We therefore anticipated that the effects of CH on endoderm induction may be attributable to recapitulation of Wnt/β-catenin signalling.

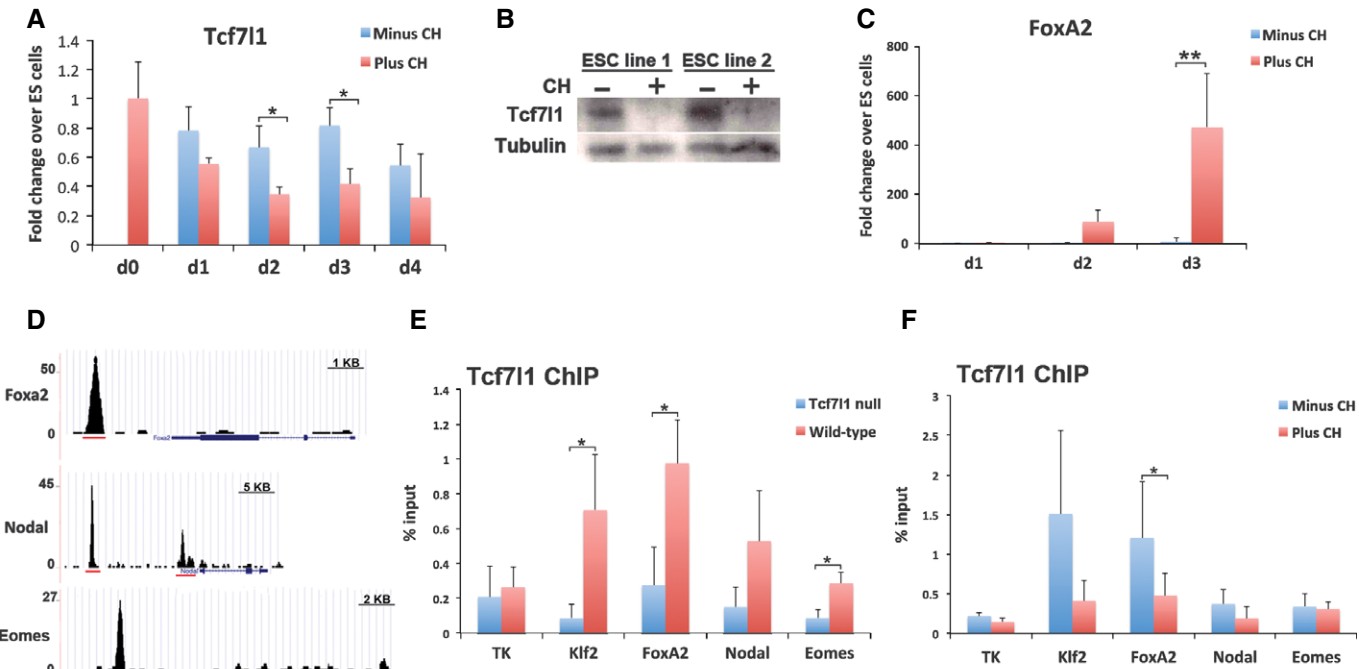

**Figure 2.  GSK3 inhibition reduces activity of the transcriptional repressor Tcf7l1.**

A Assay of *Tcf7l1* mRNA by RT–PCR for the first 4 days of differentiation in the presence or absence of 3 μM CH. Average and SD of three independent experiments. Day 2: *P = 0.048, Day 3: *P = 0.036.

B Immunoblot for Tcf7l1 at day 3 of differentiation in the presence (+) or absence (−) of 3 μM CH. Tubulin was used as a loading control.

C Assay of *FoxA2* mRNA by RT–PCR for the first 3 days of differentiation in the presence or absence of 3 μM CH. Average and SD of three independent experiments. **P = 5 × 10$^{-7}$.

D ChIP-seq data from a compendium of transcription factor ChIP-seq analyses (Sánchez-Castillo *et al*, 2015). Gene tracks represent binding of Tcf7l1 at the *FoxA2*, *Nodal* and *Eomesodermin* (*Eomes*) gene locus. Red lines indicate the regions analysed by ChIP-PCR for Tcf7l1.

E ChIP for Tcf7l1 followed by qPCR for the regions indicated in (D). Analysis was performed in wild-type ES cells in 2i culture conditions. Average and SD of three independent experiments. Klf2: *P = 0.025, FoxA2: *P = 0.050, Eomes. *P = 0.023.

F ChIP for Tcf7l1 followed by qPCR. Analysis was performed in wild-type ES cells on day 3 of differentiation in the presence or absence of 3 μM CH. Average and SD of three independent experiments. FoxA2: *P = 0.049.

Source data are available online for this figure.

Recombinant Wnt3a (200 ng/ml, in the presence of 500 ng/ml R-spondin-1) induced a similar activation of the TOPFlash reporter as 3 μM CH during endodermal differentiation (Fig 5A). However, this treatment had no influence on *Tcf7l1* mRNA expression and resulted in substantially lower induction of *FoxA2* than obtained with CH (Fig 5B). Increasing doses of Wnt3a had no greater effect. These data indicate that Wnt3a can only partially reproduce the effect of CH.

To address directly whether the canonical Wnt/β-catenin is necessary for activation of FoxA2, we used *Ctnnb1* (β-catenin) null ES cells (Wray *et al*, 2011). We found that *Ctnnb1* null cells displayed downregulation of Tcf7l1 protein and upregulation of FoxA2 in a CH-dependent manner, albeit to a lower extent that control cells (Fig 5C and D). They also showed reduced *Tcf7l1* mRNA expression in response to CH (Fig 5C). Null cells did not progress further in differentiation, in line with the requirement for β-catenin in epithelialisation (Lyashenko *et al*, 2011).

These findings suggests that function for β-catenin is more in consolidation of definitive endoderm lineage commitment than initial specification via induction of FoxA2, which may proceed independently. Consistent with this, we found that inhibition of

Wnt/β-catenin signalling using the tankyrase inhibitor XAV-939 (Huang *et al*, 2009) did not prevent the down regulation of *Tcf7l1* and had little effect on *FoxA2* expression (Fig EV3) but prevented upregulation of *Sox17*.

### GSK3 inhibition stabilises cMyc, which suppresses Tcf7l1 expression

cMyc is a well-known target of GSK3 inhibition. GSK3 phosphorylates cMyc at Thr58 and thereby triggers degradation of the protein (Sears *et al*, 2000; Yeh *et al*, 2004). GSK3 inhibition during ES cell differentiation may therefore be predicted to result in stabilisation of cMyc. We explored whether increased levels of cMyc may play a role in attenuation of Tcf7l1.

cMyc expression is low or undetectable in naïve ES cells (Ying *et al*, 2008; Marks *et al*, 2012) and has been shown to be dispensable for self-renewal (Davis *et al*, 1993; Hishida *et al*, 2011). cMyc transcription was dramatically increased > 100-fold upon induction of endodermal differentiation (Fig 6A). Total cMyc protein also increased, peaking at day 3 of differentiation (Fig 6B). Phosphorylation of cMyc at serine 62 (pSer62) is commonly associated with

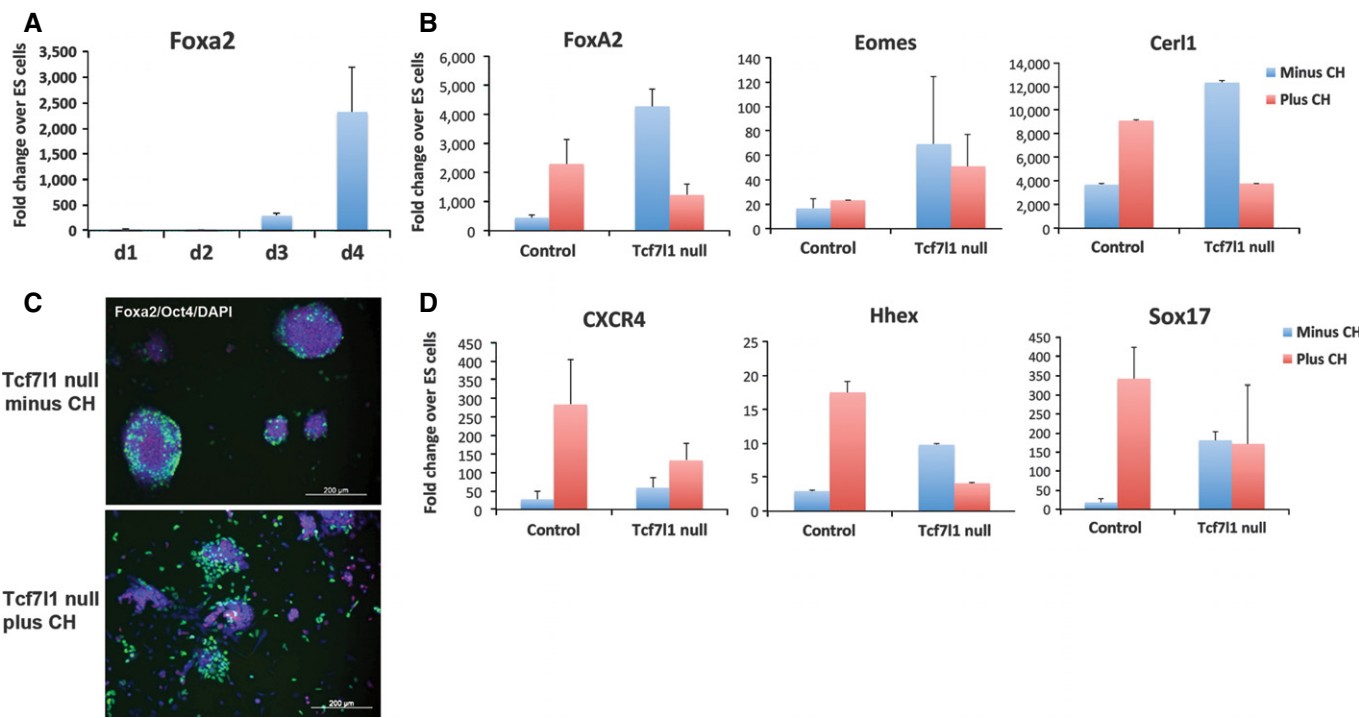

**Figure 3.  Tcf7l1 null ES cells do not require CH to initiate endoderm differentiation programme.**

A   Assay of *FoxA2* mRNA by RT–PCR in *Tcf7l1* null cells on day 1 to 4 of differentiation without CH. Average and SD of three independent experiments.
B   Gene expression analysis of endodermal associated genes by RT–PCR in control wild-type and *Tcf7l1* null cells on day 4 of differentiation in the presence and absence of 3 μM CH. Average and SD of three independent experiments.
C   FoxA2 (green) and Oct4 (red) immunostaining of *Tcf7l1* null ES cells on day 4 of differentiation in the presence or absence of 3 μM CH. DAPI staining in blue. Scale bars, 200 μm.
D   Assay of *CXCR4*, *Hhex* and *Sox17* mRNA by RT–PCR in control wild-type and *Tcf7l1* null cells on day 6 of differentiation in the presence and absence of 3 μM CH. Average and SD of three independent experiments.

stabilisation of cMyc (Sears *et al*, 2000), and levels of both pSer62 cMyc and total cMyc were increased by GSK3 inhibition with the ratio of pSer62 cMyc/total cMyc remaining constant during differentiation (Fig 6B). In contrast, the level of phosphorylation of cMyc at Thr68 (pThr58) decreased significantly in the presence of CH, with the ratio of pThr58/total protein being very low at day 2 and 3 of differentiation (Fig 6C). Without GSK3 inhibition, the ratio of pThr58/total protein was very much increased and the levels of total cMyc were lower. Similar results for total cMyc and pThr68 cMyc were observed in *Ctnnb1* null cells (Fig 6D), indicating this effect of CH on cMyc is independent of Wnt/β-catenin signalling. These data confirm that CH treatment increases stabilisation and accumulation of cMyc immediately prior to FoxA2 induction.

NMyc is closely related to cMyc, dimerises with Max, and can compensate for loss of cMyc (Lüscher & Larsson, 1999; Malynn *et al*, 2000). Since NMyc also appears to be a target of GSK3 (Kenney *et al*, 2004; Sjostrom *et al*, 2005), we examined its expression profile. NMyc transcript was detectable in naïve ES cells and increased in CH-treated cultures at day 3 of differentiation (Fig EV4).

Both cMyc and NMyc form a specific DNA-binding complex with a partner protein termed Max. Though typically associated with gene activation, Myc/Max can also repress genes by binding to their promoters in association with Miz1 (Staller *et al*, 2001; Gebhardt *et al*, 2006; Herkert & Eilers, 2010; van Riggelen *et al*, 2010; Walz

*et al*, 2014). Strikingly, pharmacological inhibitors (cMyc inhibitor II (Calbiochem) and 10058-F4 (Sigma)) that prevent the Myc/Max interaction (Zirath *et al*, 2013; Müller *et al*, 2014) significantly reduced the ability of CH to repress Tcf7l1 expression during differentiation (Fig 6E), providing evidence that the effect of CH on Tcf7l1 transcription is mediated through Myc proteins.

To confirm the requirement for Myc proteins, we analysed cells genetically deleted for cMyc. These cells also have reduced levels of nMyc following deletion of one allele ($cMyc^{-/-}$;$nMyc^{+/-}$) (manuscript submitted). In these Myc-deficient cells, the decrease in Tcf7l1 transcript we observe in response to CH in wild-type cells is lost (Fig 6F). Furthermore, FoxA2 is not induced at the transcript or protein level (Fig 6F and G). These data provide genetic evidence that Myc plays a key role in the specification of endoderm from naïve ES cells.

We then investigated whether stabilised cMyc could replace GSK3 inhibition. $cMyc^{T58A}$ is a stabilised version of cMyc insensitive to GSK3 phosphorylation (Gregory *et al*, 2003). We generated ES cell lines containing a doxycycline (Dox)-inducible $cMyc^{T58A}$ transgene (iT58A lines). *cMyc* mRNA was increased 24 h following Dox induction in a dose-dependent manner (Fig 6H). Moderate induction of $cMyc^{T58A}$ to between 5- and 100-fold the level in naïve ES cells, a similar range to that observed during early endodermal differentiation, resulted in reduced *Tcf7l1* mRNA levels (Fig 6H). Downregulation of *Tcf7l1* mRNA was lost when

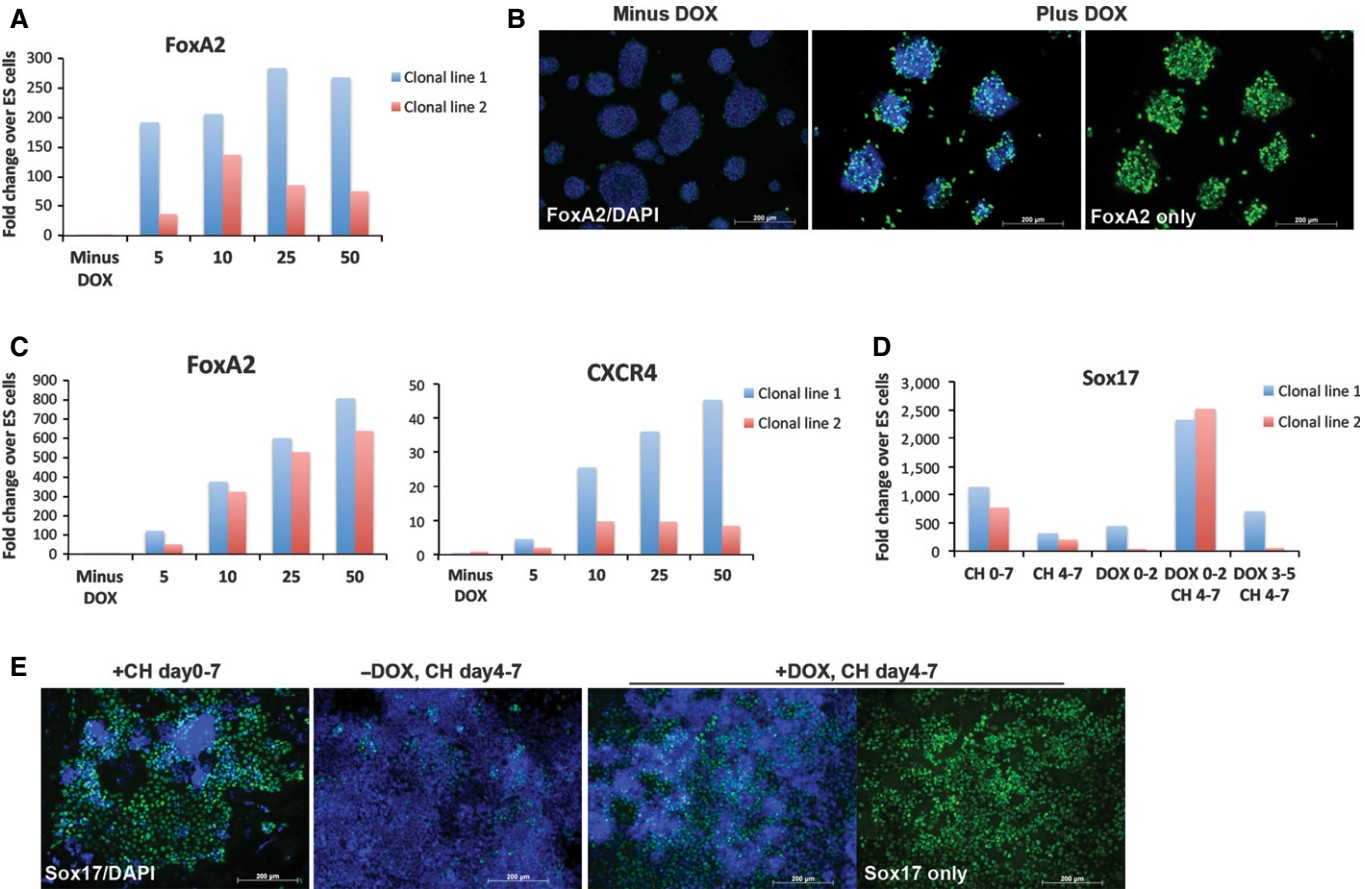

**Figure 4.  Induction of FoxA2 alleviates the requirement for CH.**

A   Assay of *FoxA2* mRNA by RT–PCR in FoxA2-Tet ES cell lines 48 h after treatment with the indicated dose of DOX. Results from two independently derived clonal lines.
B   FoxA2 immunostaining of FoxA2-Tet ES cells 48 h after DOX treatment. Left, no DOX control. Middle and right, 10 ng/ml DOX. DAPI staining shown in blue. Right panel shows middle image without DAPI signal. Scale bars, 200 μm.
C   Assay of *FoxA2* and *CXCR4* mRNA by RT–PCR on day 3 of differentiation with DOX at the indicated dose. Results from two independently derived clonal lines.
D   Assay of *Sox17* mRNA by RT–PCR on day 7 of differentiation with the addition of 3 μM CH and/or 10 ng/ml DOX at the indicated time points (in days). Results from two independently derived clonal lines.
E   Sox17 immunostaining of FoxA2-Tet ES cells at day 7 of differentiation in the presence or absence of 10 ng/ml DOX and the presence and absence of 3 μM CH as indicated. Far right panel shows image to its left without DAPI. Scale bars, 200 μm.

cMyc[T58A] was induced to very high levels (over 200 fold). Myc is involved in many cellular functions, and further complexity is brought about by its function being both dose and context dependent (Murphy *et al*, 2008). Therefore, the effects of overexpression of Myc during endoderm differentiation are uncertain since this will affect processes not relevant to the normal differentiation process. Our focus therefore was to express Myc ectopically at levels equivalent to those found during endodermal differentiation. Importantly, induction of physiological levels of cMyc[T58A] during differentiation resulted in a significant reduction in Tcf7l1 expression, without requirement for CH (Fig 6I). This was also observed in *Ctnnb1* iT58A cell lines, confirming that the effect of Myc on *Tcf7l1* expression is independent of β-catenin (Fig EV4). Tcf7l1 protein levels were also reduced in response to cMyc[T58A] induction (Fig 6J), not as pronounced as with CH but similar to observations in the *Ctnnb1*[−/−] cells in response to CH and consistent with parallel effects of GSK3 inhibition.

Collectively, these data support a model in which following CH treatment distinct effectors, Myc and β-catenin, act in parallel to ensure that Tcf7l1 function is rapidly extinguished, permitting robust activation of FoxA2 in the presence of Nodal/Activin.

### Tcf7l1 is a direct target of cMyc

The ability of cMyc[T58A] to reduce Tcf7l1 transcription suggested that the *Tcf7l1* gene could be a direct target of cMyc. The genomic region upstream of the *Tcf7l1* transcriptional start site (TSS) was examined for cMyc-binding motifs, and a cluster of 9 non-consensus cMyc-binding motifs (CATGTG or CAGGAG) was found within 3 kb of the TSS (Fig 7A).

ChIP-PCR was used to investigate cMyc binding to the *Tcf7l1* proximal promoter. Binding of cMyc to the *Tcf7l1* promoter was low in ES cells and during differentiation in the absence of CH (Figs 7B and EV5). CH induced a significant increase in binding of cMyc at

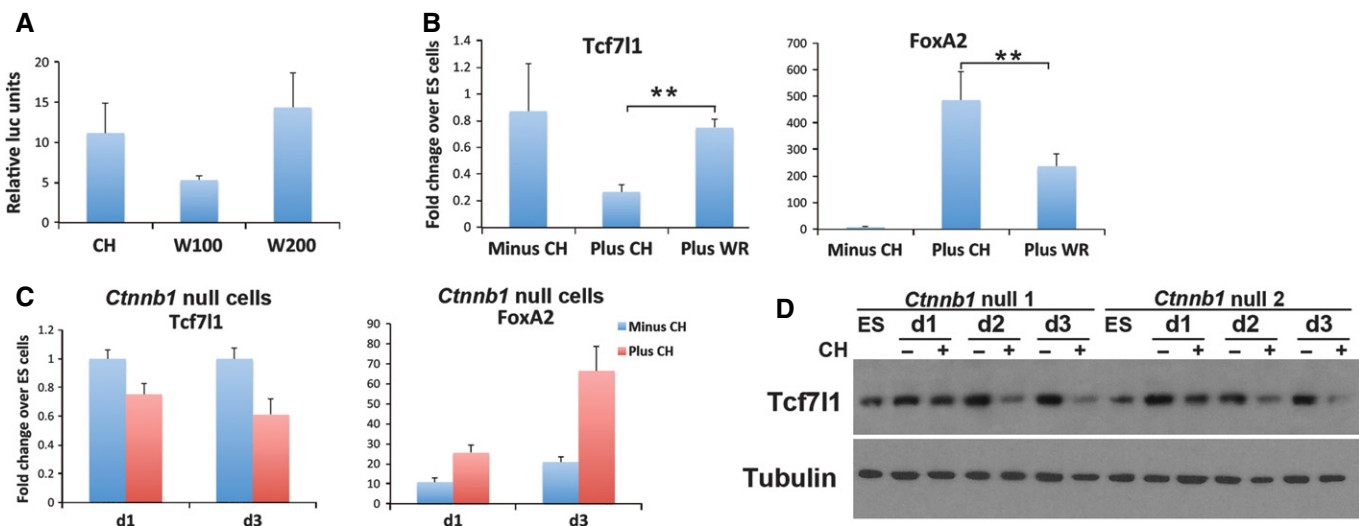

**Figure 5.  Suppression of GSK3 activity reduces Tcf7l1 and induces FoxA2 through both β-catenin-dependent and β-catenin-independent pathways.**

A   TOPFlash reporter activity following differentiation for 3 days in 3 μM CH or 100 ng/ml (W100) or 200 ng/ml (W200) recombinant Wnt3a plus 500 ng/ml recombinant R-spondin-1. Relative luciferase units are normalised to FOPFlash. Average and SD of three independent experiments.

B   Assay of *Tcf7l1* and *FoxA2* mRNA by RT–PCR at day 2 of differentiation in the presence or absence of 3 μM CH or recombinant Wnt3a (200 ng/ml plus 500 ng/ml R-spondin-1 (WR)). Average and SD of three independent experiments. **P < 0.01.

C   Assay of *Tcf7l1* and *FoxA2* mRNA by RT–PCR in *Ctnnb1* null cells at day 1 and 3 of differentiation in the absence or presence of 3 μM CH. Average and SD of three independent experiments.

D   Immunoblot for Tcf7l1 in *Ctnnb1* null cells during the first 3 days of differentiation in the absence (−) or presence (+) of 3 μM CH. Results from two independent, clonally derived *Ctnnb1* null ES cell lines are shown. Tubulin was used as a loading control.

Source data are available online for this figure.

a region in the *Tcf7l1* proximal promoter, 350 to 750 bp upstream of the TSS (region 8) (Fig 7B). cMyc binding to this region was ablated following addition of the cMyc inhibitor 10058-F4 (Fig 7C).

The potent effect of cMyc on FoxA2 expression during differentiation prompted us to investigate whether, in addition to Tcf7l1 repression, cMyc was also having a direct effect on FoxA2 expression. Several Myc binding motifs were identified in the *FoxA2* promoter, and ChIP-PCR was used to analyse cMyc binding to these regions. In contrast to the *Tcf7l1* promoter, no significant increase in binding of cMyc to the *FoxA2* promoter was observed in response to CH (Fig EV5). These data are not supportive of *FoxA2* being a direct target of cMyc. However, in the absence of genomewide ChIP data during differentiation, we cannot rule out the possibility that cMyc may also act directly on *FoxA2* and contribute to its induction when *Tcf7l1* is repressed.

cMyc/Max forms a repressive complex upon binding of Miz1 (Staller *et al*, 2001), and we identified a putative Miz1 binding motif within the cMyc binding domain in the *Tcf7l1* proximal promoter (Wolf *et al*, 2013; Barrilleaux *et al*, 2014) (Fig 7A). In addition, promoters repressed by Myc are known to be enriched for SP1 sites, which bind both Myc and Miz1 (Walz *et al*, 2014). SP1 sites are also present in the *Tcf7l1* proximal promoter around region 8 (Fig 7A).

Miz1 is expressed in ES cells and during the initial stages of endodermal differentiation (Fig 7D). ChIP-PCR was used to determine whether Miz1 binds to the *Tcf7l1* proximal promoter. Indeed, we find that Miz1 was bound to the *Tcf7l1* promoter at day 2 of differentiation and co-localised to the same region as cMyc (Fig 7E). In contrast to cMyc, Miz1 binding to this region was not dependent on

CH, suggesting that Miz1 may be pre-bound prior to recruitment of cMyc at the onset of differentiation (Fig EV5).

In summary, we find that CH-induced GSK3 inhibition during differentiation results in increased binding of cMyc to a Miz1 bound site in the *Tcf7l1* proximal promoter, and this is associated with repressed *Tcf7l1* transcription, which in turn relieves the repression of *FoxA2*.

## Discussion

In this study, we explored *in vitro* specification of naïve ES cells to definitive endoderm and deconvoluted underlying signalling and transcriptional processes. We uncovered sequential and mechanistically distinct effects of GSK3 inhibition. Downstream of GSK3 inhibition dual effector mechanisms initially converge to attenuate Tcf7l. Consequent to reduction of Tcf7l1, transcription of *FoxA2* is de-repressed. FoxA proteins are pioneer factors and have an ability to bind to and open up compacted chromatin in preparation for gene activity (Zaret & Carroll, 2011; Li *et al*, 2012). Thus, FoxA2 likely enables a cascade of downstream transcriptional and epigenetic changes that drives subsequent endoderm commitment, further facilitated by GSK3 inhibition stimulating canonical Wnt/β-catenin activity (Fig 8).

Tcf7l1 is a member of the Tcf/Lef family of transcription factors and has a well-established role in mammals as a transcriptional repressor (Merrill *et al*, 2004; Pereira *et al*, 2006; Wray *et al*, 2011; Hoffman *et al*, 2013). Genetic ablation of *Tcf7l1* leads to an expanded expression domain for FoxA2 in the mouse embryo

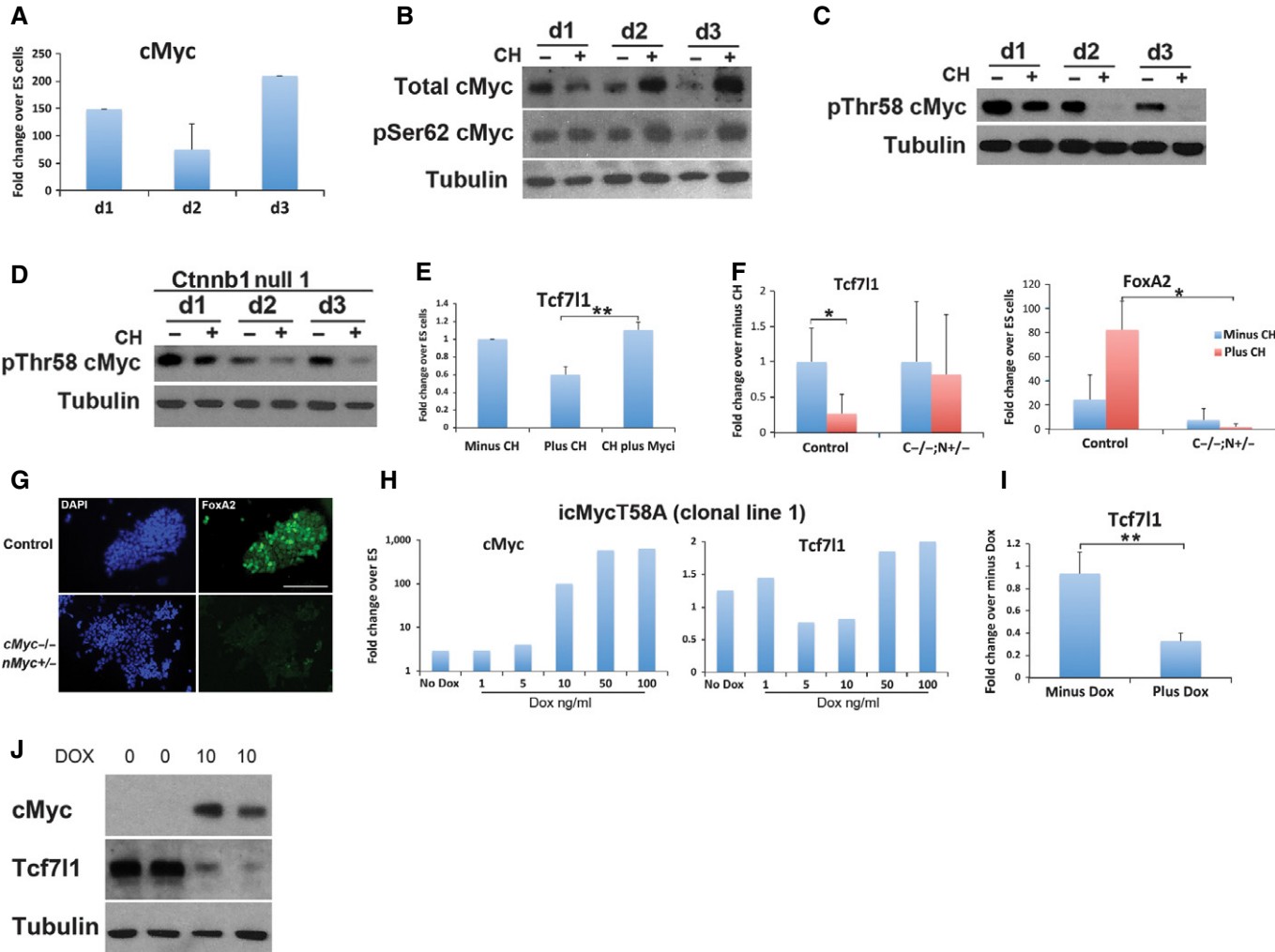

**Figure 6. cMyc protein is stabilised by GSK3 inhibition and reduces Tcf7l1 expression independently of β-catenin.**

A    Assay of *cMyc* mRNA by RT–PCR in wild-type ES cells at day 1 to 3 of differentiation. Average and SD of three independent experiments.

B    Immunoblot for total cMyc and phosphor-Ser62 (pSer62) cMyc during the first 3 days of differentiation in the absence (−) or presence (+) of 3 μM CH. Tubulin was used as a loading control.

C, D    Immunoblot for the destabilised form of cMyc protein (pThr58) during the first 3 days of differentiation in the absence (−) or presence (+) of 3 μM CH in wild-type (C) and *Ctnnb1* null (D) cells. Tubulin was used as a loading control.

E    Assay of *Tcf7l1* mRNA by RT–PCR in wild-type cells at day 2 of differentiation in the presence or absence of a small-molecule inhibitor of cMyc (cMyc inhibitor II; Myci). Treatment with Myc inhibitor 10058-F4 (Sigma) generated similar results (data not shown). Average and SD of three independent experiments. **P = 0.0002.

F    Assay of *Tcf7l1* and *FoxA2* mRNA by RT–PCR in *cMyc* null/*nMyc* heterozygous cells (C−/−;N+/−) and floxed parental cells (cMyc^fl/fl;nMyc^fl/fl; control) at day 3 of differentiation in the presence or absence of 3 μM CH. Average and SD of four or six independent experiments. Tcf7l1: *P = 0.01, FoxA2: *P = 0.034.

G    FoxA2 immunostaining of control and *cMyc* null/*nMyc* heterozygous (cMyc−/−;nMyc+/−) cells at day 3 of differentiation in the presence of 3 μM CH. Scale bar, 200 μm.

H    Assay of *cMyc* and *Tcf7l1* mRNA by RT–PCR in cMyc^T58A-inducible cell lines (icMyc^T58A) 24 h after DOX treatment. Average of two independent experiments.

I    Assay of *Tcf7l1* mRNA by RT–PCR in icMyc^T58A clone 1 cell line at day 3 of differentiation in the absence of 3 μM CH following cMyc induction with 10 ng/ml DOX. Average and SD of 3 independent experiments. **P = 0.007.

J    Immunoblot for cMyc and Tcf7l1 in icMyc^T58A clone 1 cell line 48 h after DOX induction at 10 ng/ml. Tubulin was used as a loading control.

Source data are available online for this figure.

(Merrill *et al*, 2004), consistent with our *in vitro* findings that abrogation of Tcf7l1 is a critical step in enabling activation of *FoxA2* expression.

How does GSK3 inhibition suppress Tcf7l1 activity? GSK3 is a central component in the β-catenin destruction complex and its inhibition is widely used as a mimetic of canonical Wnt pathway activation. Direct protein–protein interaction between β-catenin and

Tcf7l1 reduces DNA binding (Shy *et al*, 2013). However, we found that Wnt signalling only partially recapitulates the effects of GSK inhibition and cannot account for the prominent reduction in *Tcf7l1* mRNA. This prompted us to search for additional downstream effectors involved in endoderm specification.

GSK3 phosphorylates cMyc and targets it for proteasomal degradation. Accordingly, CH stabilises cMyc during early

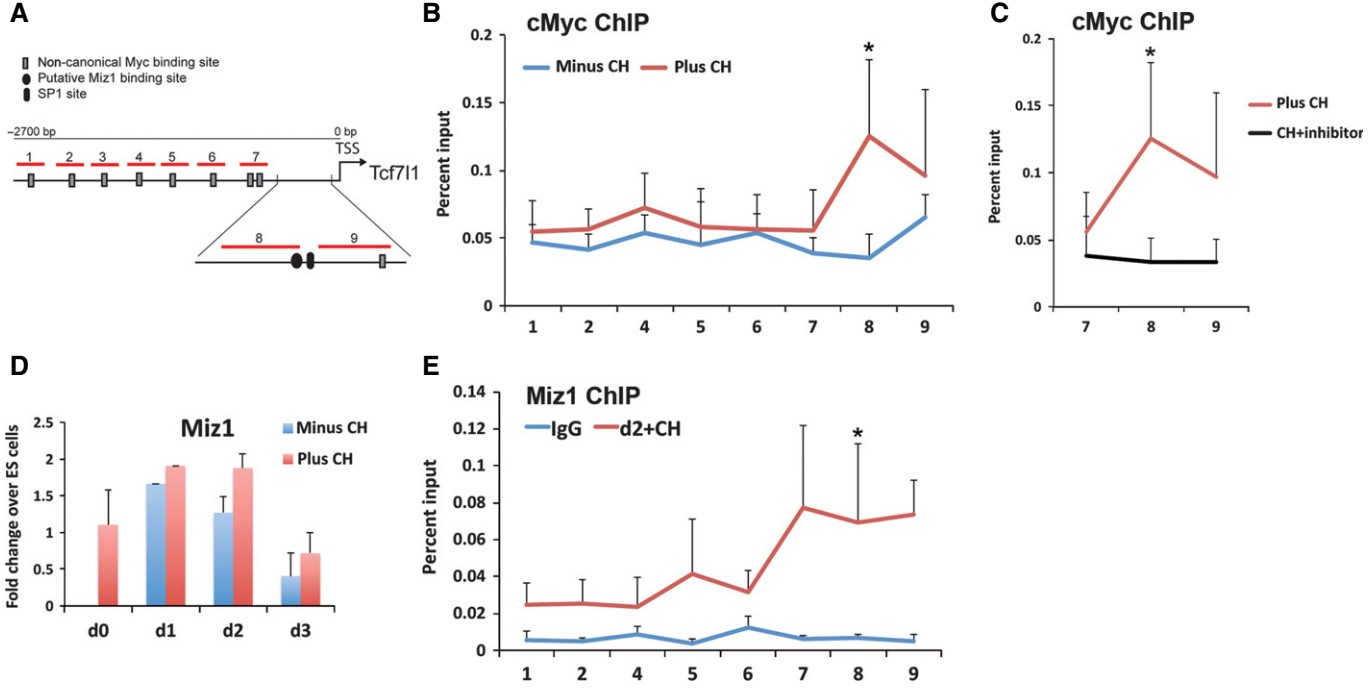

**Figure 7.  cMyc binds to the *Tcf7l1* genomic locus in response to CH.**

A   Schematic representation of the genomic region upstream of the Tcf7l1 transcriptional start site (TSS). Red lines indicate PCR amplicons designed to span the putative Myc binding sites (1–7 and 9) or putative Miz1 binding site (8).

B   ChIP for cMyc performed in wild-type ES cells differentiated for 2 days in the absence or presence of 3 μM CH. qPCR was carried out for the regions indicated in (A). Average and SD for five independent experiments, *P = 0.028.

C   ChIP for cMyc performed in wild-type ES cells differentiated for 2 days in the presence of 3 μM CH plus or minus Myc inhibitor 10058-F4. qPCR was carried out for the regions indicated in (A). Average and SD for three independent experiments, *P = 0.026.

D   Assay of *Miz1* mRNA by RT–PCR for the first 3 days of differentiation in the presence or absence of 3 μM CH. Average and SD of three independent experiments.

E   ChIP for Miz1 or IgG control performed in E14 ES cells differentiated for 2 days in presence of 3 μM CH. qPCR was carried out for the regions indicated in (A). Average and SD for five independent experiments, *P = 0.0015.

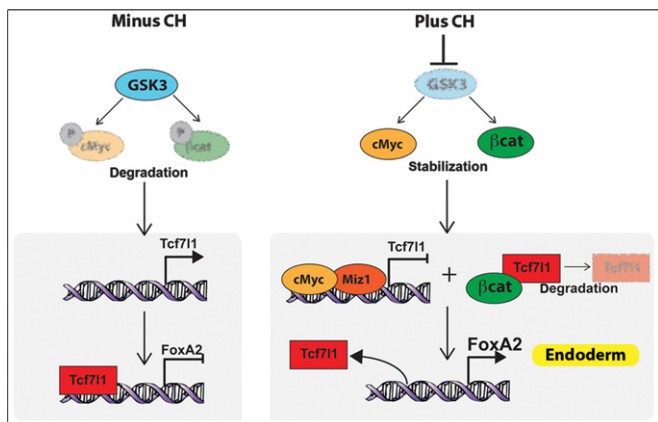

**Figure 8.   Multiple mediators of GSK3 inhibition converge on Tcf7l1 to lift the roadblock to endoderm specification.**

Inhibition of GSK3 stabilises cMyc and β-catenin protein levels. cMyc and β-catenin decrease Tcf7l1 protein levels resulting in the de-repression of *FoxA2*, allowing responsiveness to endoderm inducers such as Nodal/Activin.

endoderm differentiation. Our results demonstrate that this leads to a pronounced reduction in *Tcf7l1* expression associated with increased binding of cMyc to the *Tcf7l1* proximal promoter.

Miz1, a co-repressor with cMyc, is also significantly bound to this region suggesting Myc/Miz function together to directly repress Tcf7l1. It is also possible that cMyc has additional, indirect effects on Tcf7l1 expression, for example by activating other transcriptional or chromatin repressors. Thus, we propose that inhibition of GSK3 activity attenuates Tcf7l1 both transcriptionally (Myc repression) and post-transcriptionally (β-catenin-mediated protein degradation) (Shy *et al*, 2013). Together, these effects relieve Tcf7l1 repression of *FoxA2* to permit timely and robust endodermal specification.

ES cells cultured in the presence of serum are heterogeneous and poised for differentiation. They express cMyc at significantly higher levels than ground state ES cells (Ying *et al*, 2008; Marks *et al*, 2012). The requirement to increase the activity of cMyc at the onset of endodermal differentiation may therefore be masked in serum culture. This study starting from ES cells in a state closely resembling the pre-implantation epiblast (Boroviak *et al*, 2014) and applying defined conditions for differentiation has provided fresh insight into the requirements for endoderm induction.

Our findings also clarify the roles of Wnt/β-catenin signalling in endoderm specification. Wnt signalling certainly contributes to endoderm specification, but it is neither necessary nor sufficient to induce FoxA2 activation and initiate endodermal cell fate. Our findings indicate that the major contribution of Wnt signalling is in

induction of the later endodermal marker *Sox17* and consolidation of the endoderm gene regulatory network.

*FoxA2* transcriptional activation lies downstream of Nodal signalling (Hoodless *et al*, 2001; Xi *et al*, 2011), and consistent with this, Activin A was critical in addition to CH for efficient endodermal differentiation. Reduced levels and activity of Tcf7l1 remove a roadblock to Activin/Nodal-mediated activation of FoxA2. Thus, conjoint activity of three core effectors—cMyc, Wnt/β-catenin and Activin/Nodal—directs early endoderm induction.

Abnormal expression of TCF7L1 and FOXA2 is associated with multiple aggressive human malignancies of endodermal origin (Song *et al*, 2010; Liu *et al*, 2012; Wang *et al*, 2014). Thus, it may be of interest to explore how the molecular pathways uncovered in this study operate in human cancers. For example, does dysregulation of TCF7L1/FOXA2 lead to loss of differentiated features, a hallmark of anaplastic tumours? Related to this, FoxA2 is known to function in the balance between EMT and MET (Song *et al*, 2010).

In conclusion, our findings reveal that Tcf7l1 is a barrier to definitive endoderm specification, acting to directly block induction of the pioneer transcription factor, FoxA2. cMyc and β-catenin operate downstream of GSK3 inhibition to reduce Tcf7l1 transcription and activity, respectively. This explains the potent effects of GSK3 inhibition in promoting endoderm lineage specification from the naïve state and highlights the developmental switch in Tcf7l1 repressor function from promoting collapse of naïve pluripotency to constraining lineage segregation (Hoffman *et al*, 2013).

## Materials and Methods

### Cell lines, culture and differentiation

Mouse ES cells (E14Tg2a (Smith & Hooper, 1987) and derivatives (see below), 129/Bl6 (derived in-house), CGR8 (Mountford *et al*, 1994), NOD (Nichols *et al*, 2009), *Tcf7l1* null (Merrill *et al*, 2004; Guo *et al*, 2011), *Ctnnb1* null (Wray *et al*, 2011), HexRedStar (Morrison *et al*, 2008), cMyc$^{fl/fl}$;nMyc$^{fl/fl}$, *cMyc* null;*nMyc* het (Scognamiglio *et al*, manuscript submitted)) were cultured feeder-free on plastic coated with 0.1% gelatin (Sigma, G1890) in NDiff™ N2B27 base medium (Stem Cell Sciences Ltd, SCS-SF-NB-02) supplemented with small-molecule inhibitors PD03 (1 μM, PD0325901) and CH (3 μM, CHIR99021) (2i media). *Ctnnb1* null, cMyc$^{fl/fl}$;nMyc$^{fl/fl}$ and *cMyc* null;*nMyc* het cell lines were cultured in 2i media supplemented with 100 units/ml leukaemia inhibitory factor (LIF). Cells were re-plated every 2–3 days at a split ratio of 1 in 6 following dissociation with Accutase (PAA, L11-007).

Alternative Gsk3 inhibitors were generously gifted by Pfizer: Gsk3i1: CE-323571 (750 nM), Gsk3i2: PF-00179382 (250 nM), Gsk3i3: PF-04446208 (100 nM), Gsk3i4: PF-00179388 (100 nM) and Gsk3i5: PF-04584015 (150 nM) (Wray *et al*, 2011).

For endodermal differentiation, cells were seeded on plastic coated with 0.1% gelatin at $3 \times 10^3$ cells/cm$^2$ in NDiff™ N2B27 base medium supplemented with Activin A (20 ng/ml) (prepared in-house), Fgf4 (10 ng/ml) (R&D Systems, 235-F4), heparin (1 μg/ml) (Sigma), PI3 kinase inhibitor PI103 (100 nM) (Cayman Chemicals, 10009209) and CHIR99021 (3 μM). After 48 h, the media was changed to SF5 (DMEM:F12 plus 0.5% N2 (Invitrogen, 17502048), 1% B27 minus vitamin A (Invitrogen, 12587-010), 1 mM L-glutamine,

0.1 mM 2-mercaptoethanol (Sigma, M7522), 0.05% BSA) supplemented with Activin A (20 ng/ml), Fgf4 (10 ng/ml), heparin (1 μg/ml), EGF (10 ng/ml) (Peprotech, AF-100-15), PI103 (100 nM) and CHIR99021 (3 μM). This media was changed every 48 h for the next 5 days. Additional reagents tested in the differentiation regime: recombinant Wnt3a (R&D Systems, 1324-WN), R-spondin-1 (R&D Systems, 3474-RS), cMyc inhibitor II (cMyci) (Calbiochem, 475957), cMyc inhibitor 10058-F4 (Sigma, F3680).

Pancreatic progenitors cells (Morrison *et al*, 2008) were produced following endodermal differentiation by culturing in SF5 media supplemented with KAAD-cyclopamine (Calbiochem, 239804), 0.25 μM all-*trans* retinoic acid (Sigma, 95152) and 50 ng/ml Noggin (Peprotech, 250-38) for additional 3–5 days. Hepatic progenitors (D'Amour *et al*, 2005) were generated by culturing in NDiff™ N2B27 base medium supplemented with ascorbic acid (0.5 μM) (Sigma, A4403), monothiolglycerol ($4.5 \times 10^{-4}$ M) (Sigma, M6145), BMP4 (50 ng/ml) (prepared in-house), FGF2 (10 ng/ml), (prepared in-house), VEGF (10 ng/ml) (R+D, 293-VE), HGF (20 ng/ml) (R+D, 2207-HG-025) for additional 5 days.

### Generation of Dox-inducible cell lines

Mouse FoxA2 and cMyc open reading frame sequences were inserted downstream of a third-generation Tet-responsive promoter (P$_{TRE3G}$promoter, Clontech) using Gateway cloning (Invitrogen). The cMyc T58A variant was generated by excising a 680-bp KasI/SbfI fragment from the wild-type cMyc coding sequence and replacing it with a synthesised DNA fragment encoding a threonine to alanine change at position 58 (GeneArt Strings, Invitrogen). The P$_{TRE3G}$-FoxA2 and P$_{TRE3G}$-cMyc$^{T58A}$ constructs were used in combination with the pCAG Tet-On 3G Transactivator (Clontech) construct. Piggybac sites were included in the responder and activator constructs. Responder, activator and pBase constructs were transfected into ES cells using Lipofectamine 2000 (Invitrogen). Cells were selected on Zeocin™ (InvivoGen) and clonal lines derived. Doxycycline hydrochloride (Sigma, D3447) was added as indicated. In some cases, ES cells were maintained in serum (Sigma) and Lif (prepared in-house) for transfection and clonal expansion.

### Flow cytometry

For CXCR4 and PDGFRα analysis, cells were dissociated with PBS-based cell dissociation buffer (Gibco), resuspended in PBS/1% rat serum and incubated with APC-conjugated CXCR4 antibody or APC-conjugated PDGFRα antibody (details of antibodies used are listed in Table EV1) and analysed on a CyAn (Beckman Coulter). DAPI counterstaining was used to exclude dead cells.

### Western immunoblotting

Western blots were carried out using the following antibodies at the indicated dilutions: αTcf7l1 (Santa Cruz sc8635, 1:500), αcMyc (Santa Cruz sc42, 1:500), αp-cMycSer62 (Abcam ab185656, 1:1,000), αp-cMycThr58 (Santa Cruz sc135647, 1:500), alpha-tubulin (Abcam ab126165, 1:1,000), αgoat HRP (Abcam ab6741, 1:2,000), αmouse HRP (Abcam ab6728, 1:2,000), and αrabbit HRP (Abcam ab16284, 1:2,000).

## Chromatin immunoprecipitation

Embryonic stem cells and differentiating cells were fixed for 10 min in 1% formaldehyde, quenched with glycine, washed in ice-cold PBS and incubated for 20 min at 4°C with rotation in swelling buffer (5 mM HEPES at pH 8.25, 85 mM KCl and 0.5% NP-40). Nuclei were pelleted in a microfuge and resuspended in lysis buffer (50 mM Tris at pH 8.0, 10 mM EDTA and 1% SDS) at 4°C for 10 min. Lysates were sonicated to obtain a DNA fragment size of between 200 and 500 base pairs and diluted 1:10 in chromatin immunoprecipitation dilution buffer (50 mM Tris–HCl at pH 8.0, 167 mM NaCl, 1.1% Triton X-100 and 0.11% Na deoxycholate), pre-cleared for 2 h at 4°C with protein-G Sepharose beads (Amersham) and isotype control antibody and incubated overnight at 4°C with 2 μg Tcf3 antibody (Santa Cruz, sc8635), 2 μg Miz1 antibody (Santa Cruz H-190, sc-22837), 5 μg cMyc antibody (R&D Systems, AF3696) or control antibodies normal goat IgG (R&D Systems, AB-108-C) and normal rabbit IgG (R&D Systems, AB-105-C). Lysates were then incubated for 45 min at 4°C with blocked protein-G Sepharose beads, and beads were washed twice each in RIPA/150 mM NaCl (RIPA; 50 mM Tris–HCl at pH 8.0, 150 mM NaCl, 1 mM EDTA at pH 8.0, 1% Triton X-100, 0.1% SDS and 0.1% Na deoxycholate), taking a sample of supernatant after the first centrifugation for input DNA sample, then once in RIPA/500 mM NaCl, once in LiCl wash buffer (10 mM Tris–HCl pH 8.0, 250 mM LiCl, 1 mM EDTA) and twice in ice-cold Tris-EDTA buffer. Chromatin was eluted for 20 min at 37°C in elution buffer (10 mM Tris–HCl pH 8.0, 1 mM EDTA pH 8.0, 0.1 M NaHCO$_3$, 1% SDS). Chromatin was analysed by SYBR green real-time PCR (see Table EV2). Per cent input was calculated using standard procedures (http://www.thermofisher.com/us/en/home/life-science/epigenetics-noncoding-rna-research/chromatin-remodeling/chromatin-immunoprecipitation-chip/chip-analysis.html).

## Gene expression analysis

RNA was isolated using the RNeasy Kit (Qiagen), reverse-transcribed using SuperScript III (Life Technologies) according to the manufacturer's instructions and analysed by real-time PCR using TaqMan Fast Universal Master Mix and TaqMan probes (Applied Biosystems), the Universal Probe Library system (UPL, Roche) or SYBR green. Primers and UPL probe numbers are detailed in Table EV2. A Taqman probe endogenous control (Gapdh, glyceraldehyde-3-phosphate dehydrogenase or beta actin) was used to normalise expression.

## Immunocytochemistry

Monolayer cells were washed in PBS, fixed in 4% paraformaldehyde and permeabilised with PBS/0.1% Triton X-100. Details of antibodies used are listed in Table EV1.

## Luciferase assay

$10^5$ cells per well were transfected with 0.8 μg TOPFlash or FOPFlash (Upstate) and 0.04 μg Renilla in 24-well plates. 24 h later, cells were lysed and analysed using the dual luciferase kit (Promega) according to the manufacturer's protocol.

## Statistical analysis

Paired-sample *t*-tests (two-tailed) were used to determine significant difference between data sets of $n = 3$ or above.

**Expanded View** for this article is available online.

## Acknowledgements

We are grateful to Brad Merrill for generously providing *Tcf7l1* null ES cell lines Jason Wray and Ge Guo provided reagents. Masayo Fujiwara, Melanie Rittirsch and Rosalind Drummond provided technical assistance. Peter Humphreys and Rachael Walker provided specialist technical support. Steven Pollard provided helpful comments on the manuscript.

This study was funded by the Juvenile Diabetes Research Foundation International, the European Commission FP7 project BetaCellTherapy (agreement No. 241883), a core support grant from the Wellcome Trust and MRC to the Wellcome Trust—Medical Research Council Cambridge Stem Cell Institute, and a University of Edinburgh Chancellor's Fellowship awarded to GM. GM was a JDRF advanced postdoctoral fellow. AS is a Medical Research Council Professor.

## Author contributions

GM conceived the study and carried out, analysed and interpreted experiments. RS and AT generated the cMyc null;nMyc het mouse ES cell lines. AS supervised the study and wrote the paper with GM.

## Conflict of interest

The authors declare that they have no conflict of interest.

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
