## [Review Process File · The EMBO Journal]

Manuscript EMBO-2015-92116

Convergence of cMyc and β -catenin on Tcf7l1 enables endoderm specification

Gillian Morrison, Roberta Scognamiglio, Andreas Trumpp, and Austin Smith

Corresponding author: Gillian Morrison, University of Edinburgh

Review timeline:

Submission date:	21 May 2015
Editorial Decision:	18 June 2015
Revision received:	10 September 2015
Editorial Decision:	16 October 2015
Revision received:	03 November 2015
Accepted:	09 November 2015

Editor: Karin Dumstrei

Transaction Report:

1st Editorial Decision

18 June 2015

Thank you for submitting your manuscript to The EMBO Journal. Your manuscript has now been seen by two referees and their comments are provided below.

As you can see the referees appreciate the insights gained into endoderm differentiation in embryonic stem cells. However, they also find that the study has to be considerably strengthened along different lines. The concerns are clearly outlined below and I expect that you should be able to sort them out with additional experiments. I would therefore like to invite you to submit a suitably revised manuscript for our consideration. I should add that it is EMBO Journal policy to allow only a single round of major revision only and that it is therefore important to resolve the issues raised at this stage.

When preparing your letter of response to the referees' comments, please bear in mind that this will form part of the Review Process File, and will therefore be available online to the community. For more details on our Transparent Editorial Process, please visit our website: http://emboj.embopress.org/about#Transparent_Process

Thank you for the opportunity to consider your work for publication. I look forward to your revision.

REFeree REPORTS

Referee #1:

This is a potentially interesting paper that further delineates pathways that specify endoderm differentiation in embryonic stem cells. The authors suggest that Tcf711 is a critical repressor of FoxA2, a master regulator of endoderm differentiation, and that a Myc/Miz1 complex represses Tcf711 when Gsk3 is inhibited, allowing endoderm to form. Much of the data is convincing. In contrast, the experiments addressing the role of Myc are not yet.

Comments

1. T58P-Myc is not per se an unstable form of Myc. For example, in neuroblastoma cells virtually all of N-Myc is phosphorylated at T58. But N-Myc is also phosphorylated at S62 and the doubly-phosphorylated form is active and fairly stable. Antibodies are available for both sites and the authors really need to plot total Myc and both phosphorylated forms from the relevant experiments and show the ratio T58P/total Myc to draw any conclusions. It is worth mentioning that in the vast majority of experiments, there is no significant difference between total Myc and T58P-Myc levels so the effects shown in Figure 6B/C are very surprising.
2. In Figure 6B, the differences in Myc levels between apparent replicas are as large or larger as the differences between experimental conditions (see d2 in particular). This needs to be repaired.
3. In Figure 6H, the very moderate repression by Myc is relieved when higher levels of Myc are expressed. This is unclear and the reference given (Cartwright) does not address the issue. In particular, the Myc levels at 50 or 100nM correspond to levels that do not appear to be significantly higher than seen for endogenous Myc.
4. It is open whether the very moderate effects of Myc on Tcf711 mRNA levels translate into significant changes in Tcf711 protein levels or, as the model implies, changes in endoderm differentiation. Some functional data are required.
5. The authors implicate a Myc/Miz1 complex in repression. This is possible, although this complex so far has only been seen in transformed, not in primary cells; the authors data suggest that this difference is due to suppression of Gsk3 activity in transformed cells. A reliable way to test the involvement of Miz1 functionally is by expressing a point mutant of Myc (MycV394D) that has strongly decreased affinity to Miz1. The authors need to test this mutant for repression.

Referee #2:

Here, Morrison and Smith report a nuanced study of how suppression of GSK3 signaling promotes the generation of endoderm from naïve mouse embryonic stem cells (mESCs), principally acting via the transcriptional regulators Tcf711 (formerly known as Tcf3) and c-Myc. Starting from "2i"-grown ESCs, the authors devise a culture regimen to differentiate them into endoderm cells, involving continuous activation of Activin/Nodal (Activin A) and Fgf (Fgf4 and heparin) pathways and inhibition of PI3K signaling (PI103) for 1 week (with Egf also added for the last several days). The authors show differentiation is significantly augmented if small-molecule GSK3 inhibitors are additionally provided throughout this entire time period (Fig. 1 and Extended Fig. 1). The authors then highlight a molecular cascade through which GSK3 inhibition promotes endoderm formation. It is known that GSK3 phosphorylates and destabilizes c-Myc, and thus GSK3 inhibition leads to c-Myc accumulation (Fig. 6b). They show that c-Myc subsequently suppresses the expression of transcriptional repressor Tcf711, probably by binding to a site in the Tcf711 promoter element (Fig. 7c). Loss of Tcf711 then leads to de-repression of the key endoderm gene Foxa2. This overall is plausible, as it would parallel what happens in vivo where loss of Tcf711 expression precedes the formation of primitive streak/definitive endoderm in the mouse embryo (Hoffman et al., 2013).

While this is a rigorous mechanistic study that has revealed a previously unreported mechanism by which extrinsic signaling leads to repression of Tcf711 to enable endoderm specification, the present manuscript might be strengthened by taking into account several important issues.

1. What exact developmental transition(s) does the GSK3 inhibitor promote?

The transition from naïve pluripotency towards definitive endoderm goes through a series of (at least) three distinct, quantal steps, from naïve pluripotency to primed pluripotency to primitive streak and then into definitive endoderm. This ordered series of developmental steps is important to take into account when considering the authors' claim that GSK3 inhibition promoted endodermal lineage specification, as it is unclear which stage of differentiation was benefited (or even if any steps were inhibited) by continuous GSK3 inhibition during the prolonged 1-week course of inhibitor treatment. As the authors report, the inclusion of CHIR99021 in their differentiation system led to a "reproducible increase in proportion of Cxcr4+ and Sox17+ cells from <15% to ~80%" (pg. 4). Yet, this is seemingly in contrast to previously work by the authors' group as well as others (Ying et al., 2008; Nature; Wray et al., 2011; Nat Cell Biol; ten Berge et al., 2011; Nat Cell Biol) have shown that GSK3 inhibition robustly enhances the maintenance of naïve pluripotency and blocks differentiation towards primed pluripotency (the epiblast stem cell [EpiSC]-like state). Therefore one might initially anticipate that continuous GSK3 inhibition for 1 week of differentiation might actually help sustain undifferentiated naïve ESCs and therefore strongly restrict endoderm formation (at the first step of differentiation). On the other hand, there is stronger evidence that for the second step of differentiation (from primed pluripotency to the primitive streak) that Wnt activation/GSK3 inhibition promotes primitive streak formation, as the authors have cited for in vitro studies and is known in vivo (Liu et al., 1999; Nat Genetics). However, in the third step from the primitive streak towards definitive endoderm, it is generally thought that GSK3 inhibition actually antagonizes endoderm formation.

Thus, since the GSK3 inhibitor (CHIR99201) was added throughout the entire span of differentiation in the present study, it remains unclear if CHIR99201 indeed broadly promoted "endodermal" differentiation of these cells throughout all of these stages, or just over a brief defined window (for instance, for PS formation). Thus a timecourse of the different stages of differentiation in the authors' system to reveal when different successive lineages emerge would be helpful to clarify this issue, especially as the authors have already previously developed a Rex1-GFPd2 reporter cell line to sensitively track the transition from naïve to primed pluripotency (Wray et al., 2011). For instance, does CHIR treatment actually initially sustain the Rex1+ naïve cell population at the first few days of differentiation? After establishing the temporal emergence of different developing lineages at distinct steps of differentiation, a timecourse of pulsed CHIR treatment to see what exact developmental stage/timeframe at which it benefits endoderm induction would be useful.

The same general temporal caveat applies to analyses of Tcf711 function, as the authors mainly utilized Tcf711^{-/-} mESCs in which the gene was permanently removed, and thus its temporal effects are not well worked out in the current study; it is unclear whether it affected the earlier transition between mouse ESCs to mouse EpiSC-like stage, or later transitions to primitive streak and endoderm. Indeed, since absence of Tcf711 is known to trap cells in naïve pluripotency and overall hindering differentiation (Pereira et al., 2006 and Tam et al., 2008), this probably explains the authors' contradictory findings why they find that Tcf711 deletion leads to delayed expression of Foxa2 (Fig. 3b), even though their overall conclusion is that Tcf711 removal should ultimately enhance Foxa2 expression.

2. Lineage markers

Secondly, while the authors focus on the generation of definitive endoderm in their study, the markers that the authors examined are broad endoderm markers and thus it is unclear whether definitive endoderm (embryonic endoderm) or primitive endoderm (extraembryonic endoderm) is being generated through their approach. While endodermal markers such as Sox17, Foxa2 and Cxcr4 are appropriately assessed by authors, it should be noted that they are not specific to definitive endoderm cells, and each of these markers is known to be present in primitive endoderm as well (Kanai-Azuma et al., 2002; Development; Drukker et al., 2012; Nat Biotechnol amongst other reports). The authors' group, as well as others, have shown the naïve pluripotent cells are certainly capable of developing into primitive endoderm (Marks et al., 2012; Cell) so there is an impetus to ascertain whether the authors' endoderm is primitive or definitive in lineage. Finally, though the authors mention that ~80% of cells are Cxcr4+, it remains unclear what the remaining 20% of cells are, though the authors suggest that they are likely not either pluripotent or mesodermal (Fig. 1b).

Additionally, while the premise of the study is that mouse ESCs were differentiated towards endodermal cells, CHIR (and Wnt) treatment has been widely used to induce mesoderm differentiation in a number of studies. Hence, the continuous CHIR treatment used to induce endoderm presented here is somewhat contrary to prior expectations. Although gene expression of mesodermal markers such as Tbx6 and Flk1 were examined, more mesodermal markers (as well as ectodermal markers) should be examined.

Finally, on the matter of lineage markers, the authors might improve the clarity of their figures, as in their current state, some figures and figure legends are not labeled and are difficult to interpret. In Fig. 3b,d it is unclear whether red and blue colors in the histograms refer to either plus or minus CHIR conditions, and neither the figure nor the legends are labeled. In the preceding figure (Fig. 2), red alternately refers to either plus or minus CHIR in different subpanels, and the same applies for blue shading as well.

3. Myc

The authors' finding that c-Myc represses Tcf711, and thus by extrapolation might help induce endoderm formation, is interesting and is a strength of their study. However, the authors focus mainly on c-Myc blocking Tcf711 expression (Fig. 6,7) and thus since their main message is on endoderm differentiation, this raises the question of whether c-Myc (beyond repressing Tcf711) directly promotes endoderm formation, which the authors did not functionally show. Indeed this pro-endoderm role for Myc would be unexpected, as it was previously shown genetically that Myc blocks the expression of endoderm regulator Gata6 (Smith et al., 2010; Cell Stem Cell). For instance, does Myc overexpression increase expression of endodermal genes in the authors' differentiation system? Moreover, if GSK3 inhibition decreases Tcf711 by stabilizing c-Myc (which the authors suggest is the case through small molecule regulators of Myc interaction; Fig. 6e), this raises the question of whether c-Myc is truly critical for the GSK3 inhibitor's effect. Namely, can c-Myc mESCs (which have been previously described) can differentiate into endoderm?

In summary, this is a nuanced mechanistic interrogation of how GSK3 inhibition promotes differentiation from naïve pluripotency into endoderm, although this study could be improved by examining the exact developmental step that GSK3 inhibition promotes and further analyses of definitive vs. primitive endoderm lineage markers, amongst other points.

Minor points

1. The phrase "CXCR4+/Sox17+ endoderm" suggests coexpression of these markers, whilst Figure 1A depicts individual staining for Cxcr4+ or Sox17+ cells but not costaining for both simultaneously (Page 4)
2. Extended Fig. 1c: In the FACS plots, only the horizontal axis (Cxcr4) is labeled, but what is being displayed on the vertical axis?

1st Revision - authors' response

10 September 2015

We appreciate the constructive comments of the referees and have followed their suggestions. In general the referees expressed some concern about the significance of the biochemical studies of Myc. We therefore took an independent approach and analysed ES cells deficient in Myc (*cMyc*^{-/-}; *Nmyc*^{+/-}). These data, presented in Figure panels 6F and 6G, provide compelling genetic evidence of a key role for Myc in Tcf711 repression and subsequent induction of FoxA2.

Below is a point by point response to the specific issues raised by the referees.

Referee #1:

Point 1

T58P-Myc is not per se an unstable form of Myc. For example, in neuroblastoma cells virtually all of N-Myc is phosphorylated at T58. But N-Myc is also phosphorylated at S62 and the doubly-phosphorylated form is active and fairly stable. Antibodies are available for both sites and the authors really need to plot total Myc and both phosphorylated forms from the relevant experiments and show the ratio T58P/total Myc to draw any conclusions. It is worth mentioning that in the vast majority of experiments, there is no significant difference between total Myc and T58P-Myc levels so the effects shown in Figure 6B/C are very surprising.

Response to point 1:

We performed Western blots to analyse total cMyc and S62P cMyc during differentiation and include the new data as Figure 6B complementing the T58P cMyc immunoblot in Figure 6C. These results show an increase in total cMyc in response to CH and a parallel increase in S62P cMyc while levels of T58P cMyc are greatly reduced.

This result is consistent with published data in other cell types that GSK3 inhibition results in an increase in stabilized cMyc by reducing the amount of cMyc phosphorylated at position T58.

Point 2:

In Figure 6B, the differences in Myc levels between apparent replicas are as large or larger as the differences between experimental conditions (see d2 in particular). This needs to be repaired.

Response to point 2:

We agree that there was a high degree of variation between the independent samples presented in the previous total cMyc immunoblot (Fig 6B), but the trend was the consistent. We have now repeated this experiment and again and the new data in figure 6B in conjunction with S62P staining confirm the clear increase in cMyc in response to CH (peak day 3). These data are now more conclusive.

Point 3:

In Figure 6H, the very moderate repression by Myc is relieved when higher levels of Myc are expressed. This is unclear and the reference given (Cartwright) does not address the issue. In particular, the Myc levels at 50 or 100nM correspond to levels that do not appear to be significantly higher than seen for endogenous Myc.

Response to point 3:

The repression of Tcf711 is indeed lost when cMyc is highly overexpressed. Since Myc can act genome-wide, effects of massive overexpression are unpredictable and likely to be non-physiological.

The level of cMyc induction shown in the original Fig 6 was a substantial underestimate because the data were normalised to non-induced samples in the same clonal line containing the transgene. Baseline cMyc expression is increased in the inducible cell lines compared to wild-type ES cell lines, most likely due to a low level of “leaky” expression from the transgene. On consideration of the reviewers point we now present the cMyc levels as normalized to wild-type ES cell levels. This shows fold changes of 102 fold for 10 ng/ml DOX, 600 fold for 50 ng/ml DOX and 675 fold for 100 ng/ml DOX. Endogenous cMyc levels during differentiation are approximately 100 fold increased over naive ES cell levels. Thus the levels of cMyc induced at 50 and 100ng/ml DOX represent substantial overexpression.

Point 4:

It is open whether the very moderate effects of Myc on Tcf711 mRNA levels translate into significant changes in Tcf711 protein levels or, as the model implies, changes in endoderm differentiation. Some functional data are required.

Response to point 4:

We have now included data to show the effects of cMyc induction on Tcf711 protein levels. Figure 6J shows a clear reduction in Tcf711 protein in response to increased cMyc (in the absence of CH). The reduction in Tcf711 protein is not as pronounced as with CH, supporting our model that both Myc and b-catenin are necessary to reduce Tcf711 protein levels sufficiently for endoderm differentiation.

Most importantly we have now performed a genetic test of the requirement for Myc. We carried out a series of experiments using cMyc null/NMyc heterozygous (*cMyc*^{-/-};*nMyc*^{+/-}) ES cells that have been generated in the laboratory of our collaborator Prof Andreas Trumpp. We found that in the

absence of cMyc repression of Tcf711 in response to GSK3 inhibition was lost. Consequently FoxA2 was not induced at the transcript or protein level. These results, shown in Figure 6 F and G, have strengthened and confirmed our findings of a role for Myc in endoderm specification.

Point 5:

The authors implicate a Myc/Miz1 complex in repression. This is possible, although this complex so far has only been seen in transformed, not in primary cells; the authors data suggest that this difference is due to suppression of Gsk3 activity in transformed cells. A reliable way to test the involvement of Miz1 functionally is by expressing a point mutant of Myc (MycV394D) that has strongly decreased affinity to Miz1. The authors need to test this mutant for repression.

Response to point 5:

We followed the referee's suggestion and generated a set of new ES cells lines containing a DOX-inducible cMycV394D expression cassette using theTet3G system. Induction of the mutant cMyc caused less reduction in Tcf711 expression than wild-type cMyc but the results were variable and the differences not statistically significant. The experiment is challenging because we cannot control well the expression levels of cMycV394D and in addition it has some toxic effect on ES cells. The latter leads to apparent underloading in immunoblotting analyses. We have elected not to include these results in the paper because although the trend is in favour of a cMyc/Miz1 interaction, the data are not conclusive. For the referee's information we provide as an addendum to this response panels showing the RT-PCR and immunoblot analyses (page 8).

Referee #2:

Point 1

What exact developmental transition(s) does the GSK3 inhibitor promote?

.....[since] the GSK3 inhibitor (CHIR99201) was added throughout the entire span of differentiation [it is unclear] if CHIR99201 indeed broadly promoted "endodermal" differentiation of these cells throughout all of these stages, or just over a brief defined window (for instance, for PS formation). Thus a timecourse of the different stages of differentiation in the authors' system to reveal when different successive lineages emerge would be helpful.....

.....does CHIR treatment actually initially sustain the Rex1+ naïve cell population at the first few days of differentiation? After establishing the temporal emergence of different developing lineages at distinct steps of differentiation, a timecourse of pulsed CHIR treatment to see what exact developmental stage/timeframe at which it benefits endoderm induction would be useful.

Response to point 1:

We agree that it is important to delineate which developmental transitions are promoted by GSK3 inhibition and we have now included data in our revised manuscript from time-course experiments performed during differentiation. Expression of the primitive streak markers Tbra and Wnt3 are promoted by GSK3 inhibition. However, the most striking effect of GSK3 inhibition is on the induction of the key definitive endoderm markers FoxA2 and Sox17.

We conclude from these data that GSK3 inhibition initially operates on both the induction of early primitive-streak genes and on definitive endoderm genes. These data are now presented in supplementary figure EV1B.

As suggested, we have also performed the differentiation protocol with "pulses" of CH treatment. This revealed that induction of early primitive streak markers by 2 days of CH treatment was not sufficient to induce FoxA2 expression. In addition, removal of CH following induction of FoxA2 at day 3 resulted in loss of Sox17 expression and endoderm. This is consistent with our previous finding that continuous GSK3 inhibition is a requirement for induction of Sox17 (Figure 4). These new data are presented in supplementary figure EV1 (C).

With regards to an effect of CH on maintenance of naïve pluripotency, it is important to note that

differentiation conditions include Activin, FGF, and PI3K inhibitor in addition to CH. In these conditions CH delays but does not prevent down-regulation of the critical naïve pluripotency associated gene *Nanog* (now shown in Fig EV1C). Since *Nanog* expression is also regulated by Activin, potentially complicating this issue, we analysed an additional marker of naïve pluripotency, *Klf2* and found it to be efficiently down-regulated in the presence of CH (Fig EV1C).

The same general temporal caveat applies to analyses of Tcf711 function..... as the authors mainly utilized Tcf711-/- mESCs in which the gene was permanently removed, and thus its temporal effects are not well worked out in the current study; it is unclear whether it affected the earlier transition between mouse ESCs to mouse EpiSC-like stage, or later transitions to primitive streak and endoderm.

Indeed, since absence of Tcf711 is known to trap cells in naïve pluripotency and overall hindering differentiation (Pereira et al., 2006 and Tam et al., 2008), this probably explains the authors' contradictory findings why they find that Tcf711 deletion leads to delayed expression of Foxa2 (Fig. 3b), even though their overall conclusion is that Tcf711 removal should ultimately enhance Foxa2 expression.

Response to point 1 (continued):

The referee is correct that GSK3 inhibition and *Tcf711* have different effects depending on differentiation stage and signaling context. In this study we explored the role of GSK3 inhibition in the context of active MEK/ERK and Activin/Nodal signaling. This is different from the studies referenced by the reviewer where the effect of *Tcf711* deletion is studied in the absence of a strong inductive signal such as Activin.

Tcf711 null cells are not trapped in a pluripotent state in these conditions. We stated explicitly that differentiation is delayed in *Tcf711* null cells, consistent with raised levels of *Nanog* and *Klf4* in the ES cell state. Importantly, however, *FoxA2* is subsequently fully induced with a delay of only 24 hours. There is no contradiction in these findings, which reflect alternative targets and functions of *Tcf711*. We have added a sentence to the discussion to make the switch in function of *Tcf711* explicit (and consistent with previous reports, e.g. Hoffman et al). Please note that the key result for the present study is that up-regulation of *FoxA2* occurs in *Tcf711* null cells during differentiation without requirement for GSK3 inhibition.

Point 2. Lineage markers

..... the markers that the authors examined are broad endoderm markers and thus it is unclear whether definitive endoderm (embryonic endoderm) or primitive endoderm (extraembryonic endoderm) is being generated.....

*Finally, though the authors mention that ~80% of cells are *Cxcr4*⁺, it remains unclear what the remaining 20% of cells are, though the authors suggest that they are likely not either pluripotent or mesodermal (Fig. 1b).*

*Additionally, while the premise of the study is that mouse ESCs were differentiated towards endodermal cells, CHIR (and *Wnt*) treatment has been widely used to induce mesoderm differentiation in a number of studies. Hence, the continuous CHIR treatment used to induce endoderm presented here is somewhat contrary to prior expectations. Although gene expression of mesodermal markers such as *Tbx6* and *Flk1* were examined, more mesodermal markers (as well as ectodermal markers) should be examined.*

Finally, on the matter of lineage markers, the authors might improve the clarity of their figures, as in their current state, some figures and figure legends are not labeled and are difficult to interpret. In Fig. 3b,d it is unclear whether red and blue colors in the histograms refer to either plus or minus CHIR conditions, and neither the figure nor the legends are labeled. In the preceding figure (Fig. 2), red alternately refers to either plus or minus CHIR in different subpanels, and the same applies for blue shading as well.

Response to point 2:

The pattern of gene expression we observe is in accordance with embryonic development not extraembryonic. *Brachyury*, an early primitive streak marker not associated with extraembryonic endoderm, is significantly expressed during the first few days of differentiation prior to *FoxA2*,

CXCR4 and Sox17 expression (Figure EV1B). However, we agree that additional markers will strengthen our conclusion. Therefore, we examined markers associated with extraembryonic endoderm including Sox7, PDGFR α and AFP (alpha-fetal protein) and now include these data in supplementary figure EV1D. We did not detect expression of these additional markers during the endoderm specification protocol (AFP was detected only during the extended hepatic protocol).

Immunostaining indicates that at day 7 <5% of cells are positive for Oct4 (i.e. these cells have failed or are delayed in their differentiation). Flow cytometry analysis for PDGFR α indicates ~5–10 % of cells are positive for this marker and negative for CXCR4; i.e. they are not endoderm and may be mesodermal or extraembryonic endoderm-like cells. The cells that are CXCR4 negative at day 7 may well have matured further. Thus, at day 7 of differentiation, the minor population of CXCR4 negative cells are made up of a mixed population that includes: cells that are delayed in differentiation; those differentiated towards mesoderm or extraembryonic endoderm; and cells that have matured further along the endodermal lineage and no longer express CXCR4.

We extended analysis of mesoderm induction to five independent markers, none of which are appreciably expressed. We are therefore confident that mesodermal cells are not a major product of this protocol. We have observed that high levels of GSK3 inhibition can indeed promote mesodermal differentiation of ES cells, as reported by others, but importantly, the combination and doses of GSK3 inhibitor and Activin used in our protocol directs differentiation towards the endodermal lineage and very few mesoderm-like cells are evident.

The errors in Figure 3 have been remedied. Some legends were omitted during the final preparation of these figures. We have now corrected this and the plus and minus CH panels in the bar charts have been standardized throughout the manuscript.

Point 3. Myc

The authors' finding that c-Myc represses Tcf711, and thus by extrapolation might help induce endoderm formation, is interesting and is a strength of their study. However, the authors focus mainly on c-Myc blocking Tcf711 expression (Fig. 6,7) and thus since their main message is on endoderm differentiation, this raises the question of whether c-Myc (beyond repressing Tcf711) directly promotes endoderm formation, which the authors did not functionally show. Indeed this pro-endoderm role for Myc would be unexpected, as it was previously shown genetically that Myc blocks the expression of endoderm regulator Gata6 (Smith et al., 2010; Cell Stem Cell). For instance, does Myc overexpression increase expression of endodermal genes in the authors' differentiation system? Moreover, if GSK3 inhibition decreases Tcf711 by stabilizing c-Myc (which the authors suggest is the case through small molecule regulators of Myc interaction; Fig. 6e), this raises the question of whether c-Myc is truly critical for the GSK3 inhibitor's effect. Namely, can c-Myc null mESCs (which have been previously described) can differentiate into endoderm?

Response to point 3:

The study on Gata6 by Smith et al is in the context of extraembryonic not definitive endoderm.

We have tackled the role of Myc genetically using null cells, as suggested. We find that GSK3 inhibition does not promote the production of endoderm in cells lacking cMyc, thus cMyc is a critical factor in the downstream effects of GSK3 inhibition on endoderm. See response to reviewer 1 and new Figure 6 F and G.

Minor points

1. The phrase "CXCR4+/Sox17+ endoderm" suggests coexpression of these markers, whilst Figure 1A depicts individual staining for Cxcr4+ or Sox17+ cells but not costaining for both simultaneously (Page 4)

Response:

We have changed this phrase.

2. Extended Fig. 1c: In the FACS plots, only the horizontal axis (*Cxcr4*) is labeled, but what is being displayed on the vertical axis?

Response:

We have now included a title on the vertical axis and included details in the figure legend.

Addendum_Morrison

cMycV394D mutant cells have a reduced ability to repress Tcf711.

Method

The cMyc V394D variant was generated by excising a 653 bp EcoNI/PspOMI fragment from the wild-type cMyc coding sequence and replacing it with a synthesised DNA fragment encoding a Valine to Aspartic acid change at position 394 (GeneArt Strings, Invitrogen). The PTRE3G-cMycV394D construct were used in combination with the pCAG Tet-On 3G Transactivator (Clontech) construct. Piggybac sites were included in the responder and activator constructs. Responder, activator and pBase constructs were transfected into ES cells using Lipofectamine 2000 (Invitrogen). Cells were selected on Zeocin. Doxycycline hydrochloride (DOX) was added as indicated.

(A) Assay of cMyc mRNA by RT-PCR in cMycV394D inducible cell lines (icMycV394D) 24 hr after DOX treatment. Average of 3 independent experiments.

(B) Wild-type and icMycV394D cell line following DOX induction. No viable icMycV394D cells remained after 96 hr.

(C) Assay of Tcf711 mRNA by RT-PCR in icMycV394D cell line at day 3 of differentiation in the absence of 3 μM CH following cMyc induction with 10 ng/ml DOX. Average and SD of 3 independent experiments.

(D) Immunoblot for cMyc and Tcf711 in icMycV394D cell line 1 48 hr after DOX induction at 10ng/ml.

Thank you for sending us your revised manuscript. Your study has now been re-reviewed by referee #1. As you can see below, the referee appreciates the introduced changes and is supportive of publication here. However, the referee also notes some points that would have to be sorted out before acceptance here. As you can see below, the referee has some remaining questions regarding the model and the data provided in figure 6 and 7. In particular, the referee proposes a simpler model that a far I can see would fit with the current data set as well. I would like to ask you to sort this issue and the remaining ones in a last round of revision.

Let me know if we need to discuss anything further

REFEREE REPORTS

Referee #1:

The authors have revised their manuscript and have addressed many of the issues raised during the initial review. This also remains a very interesting manuscript. However, the data in figures 6 and 7 that are intended to invoke MYC as a repressor of Tcf7l1 that acts upstream of FoxA2 remain unconvincing.

Critical issues for this part of their model remain

- (i) that the weak transcriptional repression is lost when MYC levels increase above a certain level (Figure 6H). The authors state in their reply that the effects of high MYC levels in these cells are unpredictable, but they do not tell us why this should be so - in contrast to other cells where they are transforming.
- (ii) that the magnitude of Tcf7l1 repression varies between weak repression of mRNA levels and strong protein repression, arguing that the effect of MYC on Tcf7L1 levels is much more likely to be indirect.
- (iii) that the critical effects in the ChIP experiments in Figure 7 remain within the margin of error bars. The ChIPs also lack positive controls. It is unclear, why the authors do not check any of the published genome-wide datasets to support their claim.
- (iv) that the evidence (ChIPs) involving Miz1 remains non-convincing.

In addition, Figure 6F allows no conclusion about the effect in MYC-deficient cells and Figure 6E lacks controls that the Myc has worked.

The effects of MYC on induction of FoxA2 are very clear indeed. A much simpler model would therefore suggest that MYC activates FoxA2 directly. Some published ChIPSeq datasets indeed show a MYC peak at the FoxA2 promoter. The effects of MYC on Tcf11 protein levels could be due to any of the known crosstalks between MYC and the Wnt-pathway. I am simply lost why the authors did not test the possibility of a direct effect of MYC on FoxA2 and, if FoxA2 is a direct target of MYC in ES cells, modified their model accordingly.

We thank the referee for their careful consideration of our manuscript. We have now revised our manuscript to address the final issues raised. We include a point-by-point response below.

(i) that the weak transcriptional repression is lost when MYC levels increase above a certain level (Figure 6H). The authors state in their reply that the effects of high MYC levels in these cells are unpredictable, but they do not tell us why this should be so - in contrast to other cells where they are transforming.

We were referring specifically to the effects of high levels of Myc in cells undergoing endoderm differentiation.

Myc is involved in many cellular functions and further complexity is brought about by its function being both dose and context dependent (Murphy et al. Cancer Cell 2008). Therefore the effects of

over-expression of Myc are uncertain since this will affect processes not relevant to the normal differentiation process. Our focus therefore was to express Myc ectopically at levels equivalent to those found during endodermal differentiation. We achieved this through careful titration of Dox and assaying Myc by Western blot.

(ii) that the magnitude of Tcf711 repression varies between weak repression of mRNA levels and strong protein repression, arguing that the effect of MYC on Tcf7L1 levels is much more likely to be indirect.

We consider the reduction in Tcf711 transcription to be significant but agree with the reviewer that indirect effects of Myc, such as the activation of other transcriptional or chromatin repressors, should also be considered as possible additional mechanisms of Tcf711 repression by Myc. This is now covered in the revised discussion.

(iii) that the critical effects in the ChIP experiments in Figure 7 remain within the margin of error bars.

We performed statistical analysis on all our ChIP data (paired, 2-tailed T-test). We find the increase in binding of Myc at position 8 of the Tcf711 promoter in response to CH to be statistically significant ($p = 0.028$); the decrease in cMyc binding in response to the cMyc inhibitor is also significant ($p = 0.026$) and the binding of Miz1 to the cMyc binding region to be significant ($p = 0.0015$). The error bars represent standard deviations and reflect a certain degree of variability between the 5 biological replicates, which is typical for ChIP-PCR experiments. We have now included the p-values in the legend for Figure 7 in the revised manuscript.

(iii) (continued)The ChIPs (Figure 7) also lack positive controls. It is unclear, why the authors do not check any of the published genome-wide datasets to support their claim.

The targets of cMyc vary depending on the cellular context and level of expression, therefore we do not consider targets associated with or identified in other cells/conditions to be appropriate positive controls. Since we do not know the Myc targets in definitive endoderm there is no proven positive control. The cMyc ChIP PCR data presented in Figure 7 is from naïve ES cell cultures differentiated towards endoderm; there are no published genome-wide datasets for comparison.

(iv) that the evidence (ChIPs) involving Miz1 remains non-convincing.

The Myc/Miz1 repressive complex is well documented (Staller et al. Nat. Cell Biol. 2001; Seoane et al. Nature 2002; Walz et al. Nature 2014) and we present a model of Myc/Miz1 function based on the best fit with our current data. However, the reviewer is correct that other mechanisms may be involved; therefore we adjust our statement on the role of Miz1 in our model and discuss other possible mechanisms of Myc-dependent repression of Tcf711 (also see point ii above).

Figure 6E lacks controls that the Myc has worked.

We used two independent small molecule antagonists of Myc and generated similar results, indicating that the results obtained are specific to Myc inactivation. However, it was not clear from the manuscript that we had tested both and so we have now made this explicit in the legend for Figure 7.

Figure 6F allows no conclusion about the effect in MYC-deficient cells

The lack of statistical analysis in this figure was an oversight, and we have now added the results of a paired T-test. Adding CH to the control resulted in a statistically significant reduction in Tcf711 mRNA levels ($p=0.01$), (this was also previously shown in Figure 3). In contrast, no significant alteration of Tcf711 transcript levels was found in the cMyc null samples on addition of CH

($p=0.449$). We conclude that ablation of cMyc leads to loss of the significant repression of Tcf711 by GSK3 inhibition.

The effects of MYC on induction of FoxA2 are very clear indeed. A much simpler model would therefore suggest that MYC activates FoxA2 directly. Some published ChIPSeq datasets indeed show a MYC peak at the FoxA2 promoter. The effects of MYC on Tcf11 protein levels could be due to any of the known crosstalks between MYC and the Wnt-pathway. I am simply lost why the authors did not test the possibility of a direct effect of MYC on FoxA2 and, if FoxA2 is a direct target of MYC in ES cells, modified their model accordingly.

The reviewer suggests that Myc binds to FoxA2 promoter in ES cells. Our examination of a published dataset of whole-genome Myc ChIP-seq in mouse ES cells found no evidence of cMyc (or nMyc) binding to the FoxA2 promoter and so we had not evaluated this possibility further (Chen et al. Cell 2008). Nevertheless it is possible that Myc does bind to and activate FoxA2 following its induction during early endodermal differentiation. With this possibility in mind we analysed the mouse FoxA2 promoter and identified several clusters of canonical and non-canonical Myc binding motifs. We have now performed new ChIP PCR on differentiated cell cultures with or without CH and, in contrast to our results from analysis of the Tcf711 promoter, find no evidence that GSK3 inhibition increases cMyc binding to the FoxA2 promoter at these regions. We have included these data in supplementary data figure EV5 and a statement in the results section. Furthermore we do not see FoxA2 induction when cMyc transgene is overexpressed during differentiation in the absence of CH. These data are not supportive of FoxA2 being a direct target of cMyc. However, since we do not have genome-wide ChIP data, we cannot conclusively rule out a role for Myc in directly activating FoxA2 and we have now included a statement in the manuscript in regards to this possibility.